# Chronostratigraphic framework and provenance of the Ossa-Morena Zone Carboniferous basins (SW Iberia)

M. Francisco Pereira[1*], Cristina Gama[1], Ícaro Dias da Silva I.[2], José B. Silva[3], Mandy Hofmann[4], Ulf Linnemann[4], Andreas Gärtner[4]

1- Instituto de Ciências da Terra, Departamento de Geociências, ECT, Universidade de Évora, Apt.94, 7002-554 Évora, Portugal

2- Instituto Dom Luiz, Faculdade de Ciências da Universidade de Lisboa, Campo Grande, 1749-016 Lisboa, Portugal

3- Instituto Dom Luiz, Departamento de Geologia, Faculdade de Ciências da Universidade de Lisboa, Campo Grande, 1749-016 Lisboa, Portugal

4- Senckenberg Naturhistorische Sammlungen Dresden, Museum für Mineralogie und Geologie, Germany

Correspondence to: M. Francisco Pereira (mpereira@uevora.pt)

**Abstract.** Carboniferous siliciclastic and silicic magmatic rocks from the Santa Susana-São Cristovão and Cabrela regions contain valuable information regarding the timing of synorogenic processes in SW Iberia. In this region of the Ossa-Morena Zone (OMZ), Late Carboniferous terrigenous strata (i.e. the Santa Susana Formation) unconformably overlie Early Carboniferous marine siliciclastic deposits alternating with volcanic rocks (i.e. the Toca da Moura volcano-sedimentary complex). Lying below this intra-Carboniferous unconformity, the Toca da Moura volcano-sedimentary complex is intruded and overlain by the Baleizão porphyry. Original SHRIMP and LA-ICP-MS U-Pb zircon are presented in this paper, providing chronostratigraphic and provenance constraints, since available geochronological information is scarce and only biostratigraphic ages are currently available for the Santa Susana-São Cristovão region. Our findings and the currently-available detrital zircon ages from Paleozoic terranes of SW Iberia (Pulo do Lobo Zone- PLZ, South-Portuguese Zone- SPZ, and OMZ), were jointly analyzed using the K-S test and MDS diagrams to investigate provenance. The marine deposition is constrained to the age interval of c. 335-331 Ma (Visean) by new U-Pb data for silicic tuffs from the Toca da Moura and Cabrela volcano-sedimentary complexes. The Baleizão porphyry, intrusive in the Toca da Moura volcano-sedimentary complex, yielded a crystallization age of c. 318 Ma (Bashkirian), providing the minimum age for the overlying intra-Carboniferous unconformity. A comparison of detrital zircon populations from siliciclastic rocks of the Cabrela and Toca de Moura volcano-sedimentary complexes of the OMZ suggests

that they derived from distinct sources more closely associated with the SPZ and PLZ than the OMZ. Above the intra-Carboniferous unconformity, the Santa Susana Formation is either the result of the recycling of distinct sources located in the Laurussian-side (SPZ and PLZ) and Gondwanan-side (OMZ) of the Rheic suture zone. The best estimate of the crystallization age of a granite cobble found in a conglomerate from the Santa Susana Formation yielded c. 303 Ma (Kasimovian-Gzhelian), representing the maximum depositional age for the terrestrial strata. The intra-Carboniferous unconformity seems to represent a stratigraphic gap of approximately 12-14 Ma, providing evidence of the rapid post-accretion/collision uplift of the Variscan orogenic belt in SW Iberia (i.e. the OMZ, PLZ and SPZ).

**1. Introduction**

The Variscan orogen that extends from central Europe to Iberia was reworked through discrete Carboniferous sedimentary cycles during the Laurussia-Gondwana convergence, giving rise to the formation of marine and terrestrial basins. In SW Iberia, stratigraphic correlation has been proposed for the Carboniferous synorogenic strata found in the three main tectonostratigraphic divisions of the Variscan Orogen: the Ossa-Morena (OMZ), Pulo do Lobo (PLZ) and South Portuguese (SPZ) zones (Quesada and Oliveira, 2019, and references therein). The Carboniferous siliciclastic strata in the Santa Susana-São Cristovão and Cabrela regions (OMZ) includes fossils indicating Carboniferous to Kasimovian biostratigrapic ages (Teixeira, 1938-1940, 1941; Lemos de Sousa and Wagner, 1983; Wagner and Lemos de Sousa, 1983; Pereira et al., 2006; Machado et al., 2012; Lopes et al., 2014). In the Santa Susana-São Cristovão region, Late Carboniferous siliciclastic strata of the Santa Susana Formation unconformably overlie: i) the Baleizão volcanic-subvolcanic suite that was previously dated with whole-rock Rb-Sr isochrons (Priem et al., 1986), and ii) the early Carboniferous Toca da Moura volcano-sedimentary complex, which includes volcanic rocks that have never been dated. This intra-Carboniferous unconformity was generated as consequence of regional uplift and falling sea level, leading to a change in depositional environment from Early Carboniferous marine to Late Carboniferous terrestrial (Gonçalves and Carvalhosa, 1984; Oliveira et al., 1991; Machado et al., 2012). The provenance of the above-mentioned Carboniferous strata has been discussed based on petrographic, paleontological and detrital zircon geochronology evidence (Pereira et al., 2006; Machado et al., 2012; Lopes et al., 2014; Dinis et al., 2018).

In this paper, SHRIMP and LA-ICP-MS U-Pb analyses were performed on zircon grains from silicic volcanic, subvolcanic, and siliciclastic rocks sampled in the Santa Susana-São Cristovão and Cabrela regions (OMZ, SW Iberia). The aim of this geochronology study is to establish the chronostratigraphic framework of these Carboniferous strata and to discuss their provenance using a statistical approach (Kolmogorov-Smirnov test and Mutiscaling diagrams). Thus we pay

tribute to J.R. Martínez-Catalán, who devoted part of his career to investigating the
Carboniferous synorogenic basins of NW Iberia.

**2. Geological setting**
In SW Iberia, the tectonic limit between the OMZ (Gondwanan-side) and the PLZ and SPZ
(Laurussian-side) has been regarded as constituting the tectonically reworked suture zone of the
Rheic Ocean (Andrade, 1983; Quesada et al., 1994; Simancas et al., 2005; Díaz-Apiroz et al.,
2006; Ribeiro et al., 2007; Pereira et al., 2017a) (Fig. 1). This Paleozoic suture zone has been
defined along the Beja-Acebuches ophiolitic complex (Fonseca et al., 1999, and references
therein). The Beja-Acebuches ophiolitic complex is separated from the Beja Igneous Complex
(Jesus et al., 2007, 2016) by a strike-slip fault. Metabasalts and metagabbros (i.e. the Mombeja
unit of Andrade, 1983) from the Beja-Acebuches ophiolitic complex have been dated at c. 340-
332 Ma (U-Pb zircon; Azor et al., 2008), while in the Beja Igneous Complex gabbro and
granitic rocks are relatively older, yielding crystallization ages of c. 353-342 Ma (U-Pb zircon;
Jesus et al., 2007; Pin et al., 2008). Trace element and isotopic signatures of Beja Igneous
Complex plutonic rocks indicate crustal contamination of parental magmas deriving from a
depleted asthenospheric mantle reservoir (Santos et al., 1990; Pin et al., 2008; Jesus et al.,
2016). The plutonic rocks of the Beja Igneous Complex show well-defined intrusive contacts
with previously deformed and metamorphosed sedimentary and igneous rocks of the OMZ
basement (Rosas et al., 2008; Pin et al., 2008). The Beja Igneous Complex also includes the São
Cristovão-Alcáçovas subvolcanic complex (Gonçalves and Carvalhosa, 1984), composed of
silicic sub-volcanic and volcanic rocks (i.e. the Baleizão unit of Andrade, 1983), granophyres
and porphyries dated at c. 319 Ma (whole-rock Rb-Sr isochrons; Priem et al., 1986), associated
with diabases. Porphyry dykes are found cutting across the OMZ basement that is here formed
by Cambrian igneous rocks with c. 527 Ma (Alcáçovas gneiss, Chichorro et al., 2008) deformed
and metamorphosed in the Early Carboniferous at 340 ± 6 Ma (Pereira et al., 2009). The major
and trace element geochemistry of the Baleizão porphyries indicates a calc-alkaline rhyolitic,
rhyodacitic and andesitic composition typical of magmas produced at convergent plate
boundaries (Santos et al., 1987; Caldeira et al., 2007; Ferreira et al., 2014). The Baleizão
porphyries occur as dykes and sills (Andrade, 1927) (Figs. 3a, b), overlying (Gonçalves and
Carvalhosa, 1984) the Early Carboniferous siliciclastic and volcanic rocks of the Toca da Moura
volcano-sedimentary complex (Santos et al., 1987, and references therein) (Fig. 2).
The Toca da Moura volcano-sedimentary complex is mainly composed of pelites (i.e.
"Xistinhos"; Teixeira, 1944; Fig. 3a) and greywackes, associated with andesite-to-rhyolite
volcanic rocks (lava flow and tuffs; Figs. 3c, d, e), andesitic basalt, chert layers (Gonçalves and
Carvalhosa, 1984), and a few olistoliths of basalt and limestone. Siliciclastic rocks contain well-
preserved in-situ palynomorph assemblages of Tournaisian to Visean age and reworked
palynomorphs ranging in age from the Middle Cambrian to the Early Tournaisian (Pereira et al.,
2006; Lopes et al., 2014). Based on geochemical information, this volcanism was interpreted by
Santos et al. (1987) as deriving from calc-alkaline magma produced in a continental magmatic
arc. A stratigraphic correlation was established between the Toca da Moura volcano-
sedimentary complex and the Cabrela volcano-sedimentary complex (Pereira et al., 2006) which
is located 15 km to the NW, in the Évora Massif (Pereira et al., 2007; 2012a) (Fig. 1b). The
presence of variable-scale soft-sediment structures (i.e. slumps, intraclast conglomerates and
olistoliths) in both complexes indicates gravity-induced instability during marine sedimentation.
Detrital zircon ages of a siliciclastic rock from the Cabrela volcano-sedimentary complex
interbedded with silicic tuffs (Fig. 3f) are mainly Middle-Late Devonian (82%) and Early
Carboniferous (14%), also including a few older grains (sample OM-200 from Pereira et al.,
2012a).
The Santa Susana Formation (i.e. Santa Susana basin, Domingos et al., 1983; Quesada et al.,
1990, Oliveira et al. 1991) siliciclastic rocks that outcrop along a NNW-SSE-trending narrow
discontinuous band which is 0.1-5 km wide and 12 km long unconformably overlie the Baleizão
Porphyry and the Toca da Moura volcano-sedimentary complex (Fig. 2), forming the geological
contact between these stratigraphic units often defined by faults (Gonçalves and Carvalhosa,
1984). The Santa Susana Formation is divided into two members (Machado et al., 2012, and
references therein): i) the lower member is mainly composed of coarse-grained sandstone and
conglomerate beds (Figs. 4a, b, c, d); these conglomerates include pebbles and cobbles of silicic
porphyry, rhyolite, andesite, basalt, granite, felsic tuff, pelite, sandstone, greywacke, quartzite,
phyllite, chert, and quartz (Figs. 4e, f); ii) the upper member represents a repetitive sequence of
alternating beds of pelite and sandstone interbedded with coal seams, and few beds of
conglomerate (Fig. 2). These terrestrial deposits were most probably deposited in an
alluvial/fluvial-to-fluvial/lacustrine (floodplain lakes and/or abandoned channels with abundant
vegetation) system (Machado et al., 2012, and references therein). The plant fossils identified in
the siliciclastic rocks of the Santa Susana Formation indicate a Moscovian-Kasimovian
biostratigraphic age (Wagner and Lemos de Sousa, 1983). Pelitic beds from the Upper member
include palynomorph assemblages assigned with Kasimovian age (Machado et al., 2012).
Palynomorphs ranging in age from the middle Cambrian to the early Moscovian were also
found in siliciclastic rocks of the Santa Susana Formation sampled from a borehole at a depth of
around 400 m (Lopes et al., 2014). Detrital zircon ages from upper member sandstones (Dinis et
al., 2018) are mainly distributed over Devonian-Carboniferous (41-51%), Paleoproterozoic (23-
30%) and Ediacaran-Cryogenian (16-23%) groups, and also a few Stenian-Tonian and Archean
grains.

**3. Rational and analytical methods**
U-Pb geochronology of detrital zircon from siliciclastic rocks has been extensively used in
stratigraphic correlation studies for estimating the maximum depositional age and investigating
the provenance of sedimentary sequences (Fedo et al., 2001; Dickinson and Gehrels, 2009). The
youngest detrital zircon grains found in siliciclastic rock commonly provide useful information
about depositional age, especially in areas that experienced active volcanism during sediment
accumulation (Geherls, 2014). The maximum depositional age obtained for siliciclastic rock is
often not necessarily coincident with the biostratigraphic age as defined by key fossil
assemblages (Pereira et al., 2019). Therefore, in order to overcome any doubt about the true age
of deposition, it is desirable that volcanic rocks interstratified with fossiliferous siliciclastic
rocks should be dated (Fedo et al., 2001; Bowring et al., 2006). Furthermore, the application of
zircon U-Pb geochronology to volcano-sedimentary and sedimentary sequences that are
separated by unconformities, by means of the comparative analysis of their age populations,
may be useful for estimating time intervals and revealing changes in provenance. Volcanic
rocks that lie beneath or overlie sedimentary sequences and unconformities can provide
maximum and minimum ages, respectively. When detrital zircon geochronology is linked to the
geochronology of crosscutting younger igneous rocks, then both a maximum and minimum age
bracket for deposition can be determined (Fedo et al., 2001).
In this study, SHRIMP U-Pb analyses were performed for the first time on magmatic zircon
from two tuffs from the Toca da Moura volcano-sedimentary complex (TM-1 and SCV-2; Figs.
3c, d), one tuff from the Cabrela volcano-sedimentary complex (CBR-12), one from the
Baleizão silicic porphyry (SCV-30; Fig. 3b), and a cobble of granite (SCV-7; Fig. 4e) found in a
conglomerate from the lower member of the Santa Susana Formation. Estimations of the
crystallization age of samples SCV-2, TM-1 and CBR-12 (syndepositional volcanism), and
sample SCV-30 (post-depositional) were used to validate the Tournaisian-Visean
biostratigraphic age previously attributed to the Toca da Moura and Cabrela volcano-
sedimentary complexes based on palynlogical assemblages (Pereira et al., 2006; Lopes et al.,
2014). The presence of granite cobbles and pebbles in conglomerate layers from the lower Santa
Susana Formation indicates denudation and recycling of a crystalline basement involving
granite whose age is unknown. The dating of the granite cobble (sample SCV-7) is useful for
discussing provenance and estimating the maximum depositional age of the Santa Susana
conglomerate. In addition, LA-ICP-MS U-Pb analyses were performed on detrital zircon grains
from two samples of sandstone from the upper and lower members of the Santa Susana
Formation (samples SS-1 and SS-2, respectively; Fig. 5g, h), and a sample of pelite from the
Toca da Moura volcano-sedimentary complex (sample TM-3; Fig. 5e). This new U-Pb data is
useful for discussing provenance and determining the maximum depositional ages of the two
sedimentary sequences separated by an intra-Carboniferous unconformity. Sample locations in
the Santa Susana-São Cristovão region are indicated in Figure 2. Finally, detrital zircon grains
of siliciclastic rock from the Cabrela volcanic-sedimentary complex (sample CBR-11; Fig. 5f;
equivalent to sample OM-200 of Pereira et al. 2012a) were analyzed to test for the existence of
pre-Devonian ages. The new U-Pb results obtained in the present study are compared with
previously-reported age spectra for pre-Kasimovian siliciclastic rocks from the OMZ, PLZ and
SPZ siliciclastic sequences of SW Iberia, using statistical tools.
Zircon grains for U-Pb geochronology were selected using traditional techniques: density
separation using a wilfley table (Universidad Complutense de Madrid, Spain) and also using
granulometric separation using sieves with a mesh size of less than 500 microns, density
(panning) separation procedures, and mineral identification using a binocular lens and
preparation of epoxy resin mounts with zircon grains (Universidade de Évora, Portugal). U-Pb
measurements were obtained at IBERSIMS (Universidad de Granada, Spain) using SHRIMP,
and also at the Senckenberg Naturhistorische Sammlungen Dresden (Museum für Mineralogie
und Geologie, Germany) using a LA-ICP-MS. U-Pb measurements using SHRIMP and LA-
ICP-MS followed the procedures previously described by Dias da Silva et al. (2018) and Pereira
et al. (2012a), respectively. U-Pb results are listed in Tables S1 and S2 (Supplementary
Material). Concordia curves and weighted-average means were obtained using Isoplot 4
(Ludwig, 2003) (Figs. 6 and 7). Kernel density estimation (KDE) diagrams were produced with
90-110 % concordant $^{206}Pb/^{238}U$ ages for grains younger than 1.0 Ga, and $^{207}Pb/^{206}Pb$ ages for
older grains (for further details, see Frei and Gerdes, 2009) using IsoplotR (Vermeesch, 2018)
(Figs. 8a, b). Cathodoluminescence-imaging was performed at TU Bergakademie Freiberg
(Germany) and at IBERSIMS.
The K-S test and the MDS technique were used in conjunction to compare populations of
detrital zircon U-Pb ages obtained from the Carboniferous siliciclastic rocks of the Santa
Susana-São Cristovão region using a method designed for a recent study of the provenance of
Triassic sandstones (Gama et al., in press, and references therein). The K-S test is a non-
parametric statistical tool that has been successfully used for the comparison of two populations
of detrital zircon U-Pb ages by evaluating whether they are significantly different, i.e. indicating
whether zircon age populations correlate with a similar source or not, regardless of whether they
are of different sizes, while including at least 20 measurements (DeGraaff-Surpless et al., 2003).
The probability of the observed maximum vertical difference between the cumulative
probability curves (D-value) being unrelated to age differences between the two detrital zircon
populations is given by a P-value corresponding to a confidence interval of 95% (Barbeau Jr. et
al.; 2009; Guynn and Gehrels, 2010) (Fig. S1; supplementary material). High P-values and low
D-values indicate that the observed difference between the two detrital zircon populations may
be explained by the existence of common sources (Gama et al., 2020, and references therein).
K-S analyses were carried out using an Excel spreadsheet published on the University of
Arizona Geochronological Center website at

 https://sites.google.com/a/laserchron.org/laserchron/. The MDS technique provides a means for

the comparison of samples based on quantified pairwise comparisons of their detrital zircon
ages, and is extremely useful for visualising the degree of similarity between samples in two
dimensions, i.e. greater distances between samples represent a greater degree of dissimilarity
between points on MDS diagrams (Vermeesch, 2013; Spencer and Kirkland, 2015; Wissink et
al., 2018) (Fig. 9). MDS diagrams were produced using IsoplotR (Vermeesch, 2018).

**4. U-Pb geochronology: Results**

**4.1. Volcanic silicic rocks of the Toca da Moura and Cabrela volcano-sedimentary complexes**

Sample SCV-2 is a fine-grained banded rhyolitic tuff consisting of variable size and shape
quartz and K-feldspar phenocrysts and lithoclasts (less than 1mm in diameter) dispersed in ash
matrix (Fig. 5a). Zircon grains appear as stubby-to-elongated euhedral prisms (50-150 μm in
diameter), mostly showing oscillatory concentric zoning growing on distinct cores or as simple
crystals. There are some dark inclusions, unzoned patches and transgressive variably
luminescence embayments. A total of 44 U-Th-Pb SHRIMP analyses of 44 grains yielded U
content ranging from 262 to 628 ppm. A group of 23 grains with $^{206}$Pb/$^{238}$U ages (discordance ≤
5%) yielded a weighted mean $^{208}$Pb/$^{238}$U age of 331 ± 4 Ma (MSWD = 1.2; Fig. 6a), which
probably represents the crystallization age of tuff.
Sample TM-1 is a fine-grained banded rhyolitic tuff consisting quartz, K-feldspar and biotite
phenocrysts, flattened dark-brown pumice (i.e. fiamme) and lithoclasts (less than 1mm in
diameter) enclosed in ash matrix (Fig. 5b). The zircon population is characterized by stubby
euhedral-to-sub-euhedral small (30-100 μm in diameter) grains. Magmatic grains are either
simple with concentric zoning or composite showing variably luminescence cores with
concentric zoning, unzoned, or banded zoned. These cores are surrounded by overgrowths with
concentric zoning and are occasionally diffuse or unzoned. A total of 120 U-Th-Pb LA-ICP-MS
analyses yielded U content ranging from 87 to 4136 ppm. 28 $^{206}$Pb/$^{238}$U ages (90-110% of
concordance) yield a weighted mean $^{208}$Pb/$^{238}$U age of 341 ± 10 Ma with a very poor fit (MSWD
= 6.9; Fig. 6b), as indicated by the scattering of ages along the Concordia curve. A coherent
group of 21 grains with $^{206}$Pb/$^{238}$U ages yielded a weighted mean $^{208}$Pb/$^{238}$U age of 335 ± 6 Ma
(MSWD = 1.5; Fig. 6b), providing the best age estimate for the volcanic rock (Fig. 6b). The
youngest zircon grain (c. 302 Ma) probably experienced Pb loss. The six oldest zircon grains
present Paleoproterozoic (c. 2 Ga), Neoproterozoic (c. 715 Ma) and Devonian (c. 395-378 Ma)
ages, suggesting inheritance.
Sample CBR-12 is a fine-grained rhyolitic rock in which feldspar and quartz phenocrysts and
lithic fragments occur embedded in altered very-fine grained matrix of quartz, sericite and
chlorite, including devitrified shards. Zircon (40-150 μm in diameter) appears as stubby-to-
elongated euhedral prisms, mostly showing oscillatory concentric zoning, sometimes disturbed
by inclusions, as simple grains or as overgrowths. A few crystals show banded zoning or are
diffuse or unzoned. Thirty-two analyses were performed on this silicic volcanic rock yielding U
content ranging from 288 to 2587 ppm. Twenty-two analyses with discordance ≤ 5%, yielded a
weighted mean $^{208}$Pb/$^{238}$Th age of 335 ± 2 Ma (MSWD = 1.2; Fig. 6c). The oldest two grains
yielding $^{206}$Pb/$^{238}$U ages of c. 389 and 371 Ma are interpreted to represent xenocrysts.

**4.2. Baleizão porphyry**
Sample SCV-30 is a porphyritic rhyodacite-rhyolite consisting of quartz, plagioclase, K-
feldspar, biotite and amphibole phenocryst (less than 3mm in diameter) embedded in a fine-
grained silicic matrix (Fig. 5c). The zircon population contains grains (30-140 μm in diameter)
from subrounded subhedral to prismatic euhedral. Prisms are equant to moderately elongate
showing simple internal structure characterized by concentric and sector zoning to unzoned. A
concentric zoned or unzoned rim surrounds unzoned cores of few composite grains. A total of
37 U-Th-Pb SHRIMP analyses for sample SCV-30 yielded U content ranging from 74 to 4290
ppm.  25 analyses were obtained for zircon with discordance ≤ 5%, distributed along the
concordia curve from ca. 355 to 288 Ma, and yielded a weighted mean $^{208}$Pb/$^{238}$Th age of 315 ±
6 Ma (mean square of weighted deviates, MSWD = 12; Fig. 7a). Some of the spread observed
could be due to the presence of inheritance. The oldest 10 grains yielding $^{206}$Pb/$^{238}$U ages of c.
355-337 Ma probably represent xenocrysts derived from the Toca da Moura volcano-
sedimentary complex. Eleven grains in the age range ca. 334-311 Ma yielded a weighted mean
$^{208}$Pb/$^{238}$U age of 318 ± 2 Ma (MSWD = 0.95; Fig. 7a), which is regarded as the best estimate
for the crystallization age of subvolcanic silicic rock.

**4.3. Cobble of granite found in a conglomerate from the Santa Susana Formation**
Sample SCV-7 is a cobble (20 cm in diameter) of pinkish medium-grained granite consisting of
quartz, alkali feldspar and biotite (Fig. 5d). Most zircons are stubby and elongated subeuhedral
to euhedral prisms (80 to 150 μm in diameter). Morphologically zircon grains are mostly simple
showing concentric zoning, sector zoning to unzoned, and few are composite with irregular and
unzoned small cores surrounded by a rim with concentric zoning. 40 U-Th-Pb SHRIMP
analyses were performed on sample SCV-7 with U content ranging from 348 to 3177 ppm. Of
this total of analyses 24 U-Pb ages with discordance ≤ 5%, scattered along the concordia curve
from ca. 349 to 294 Ma, yielded a weighted mean $^{206}$Pb/$^{238}$U age of 327 ± 7 Ma (MSWD = 4;
Fig. 7b). A group of six zircon grains in the age range of c. 309-294 Ma yielded a weighted
mean $^{206}$Pb/$^{238}$U age of 303 ± 6 Ma (MSWD = 0.98; Fig. 7b), which is taken as the probable
crystallization age of the granite. The remaining 19 zircon grains yielded $^{206}$Pb/$^{238}$U ages of c.
349-326 Ma, suggesting inheritance.

**4.4. Siliciclastic rocks from the Toca da Moura and Cabrela volcano-sedimentary**

**complexes**

Sample TM-3 is a laminated poorly-sorted siltstone with quartz-rich silt layers, containing

feldspar and tourmaline grains, and lithoclasts (Fig. 5e), which are intercalated with darker

layers of clay. The zircon population is mostly characterized by stubby to elongated prismatic

small grains (less than 100 μm in diameter). It includes simple and composite zircons showing

concentric, sector and banded zoning. Of a total of 82 U-Th-Pb LA-ICP-MS analyses, with U

content ranging from 19 to 4630 ppm. 36 zircon grains yield 90-110% concordance. The

number grains of sample TM-1 is not conform to the minimum of 60-100 grains often used in

provenance studies (Vermeesch, 2004), and therefore percentages based on the proportions of

ages need to be interpreted with caution. The Paleozoic population of detrital zircon (36%)

includes Early Carboniferous (9%, c. 353, 349 and 340 Ma), Ordovician (14%, c. 476-456 Ma),

Cambrian (7%, c. 531-500 Ma) and Late Devonian (6%, c. 369 and 362 Ma) grains (Fig. 8a).

The Precambrian population (64%) is predominantly Neoproterozoic (36%; c. 983-587 Ma), but

also includes Paleoproterozoic (14%; c. 2-1.8 Ga), Mesoproterozoic (8%; c. 1.3-1 Ga) and

Archean (6%; c. 2.7-2.5 Ga) grains. The three youngest zircon grains (c. 353-340 Ma) yielded a

maximum depositional age of c. 348 Ma (Tournaisian), which is in accordance with the

sedimentary age inferred from biostratigraphic constraints (Late Tournaisian to Middle-Late

Viséan; Pereira et al., 2006; Lopes et al., 2014).

Sample CBR-11 is a fine-grained poorly-to-moderate sorted siltstone consisting predominantly

of quartz and few feldspar grains and lithoclasts enclosed in silt-clay-sized particles (Fig. 5g).

Most of zircon grains are small (less than 100 μm in diameter), euhedral to subeuhedral. They

are simple grains (short, stubby to equant prisms) with oscillatory concentric and banded

zoning, and only few are composite grains with rounded cores. Of a total of 20 U-Th-Pb LA-

ICP-MS analyses, with U content ranging from 54 to 1379 ppm, 10 grains yielded 90-110% of

concordance. Five grains are Paleozoic (Carboniferous: c. 359, 351 and 346 Ma; Cambrian: c.

514 and 511 Ma) and five are Precambrian (Paleoproterozoic: c. 2.4, 2.1 and 1.8 Ga;

Mesoproterozoic: 1 Ga; Neoproterozoic: c. 603 Ma). By combining our new data with those

from sample OM-200 (Pereira et al., 2012a) collected from the same quarry of sample CBR-11,

it was found that the detrital zircon population (CB, N = 54; Fig. 8a) is largely dominated by

Paleozoic grains (90%): Late-Middle Devonian (68%), Early Carboniferous (15%), Cambrian

(4%) and Early Devonian (2%) grains, being distinct from sample TM-3 described above (Fig.

8a). The number grains of sample CB (N=54), despite being larger than that of sample TM-1, is

not conform to the minimum of 60-100 grains, and therefore the proportions of ages obtained

also need to be interpreted with caution. The youngest zircon population (N = 5) ranging from c.

353 to 346 Ma), suggest a Tournaisian maximum depositional age which is slightly older than

the sedimentary age inferred from biostratigraphic constraints (Late Tournaisian to Middle-Late
Viséan; Pereira et al., 2006).

**4.5. Siliciclastic rocks from the Santa Susana Formation**
Sample SS-2 represents medium-to-coarse grained poorly-sorted sandstone. It is mainly
composed of lithoclasts (siltstone, mudstone, quartzite, phyllite, rhyolite, basalt) and quartz
grains, but also includes muscovite and feldspar grains (Fig. 5g). The zircon population is
mostly characterized by stubby to prismatic, subrounded to subangular, grains (120-300 μm in
diameter). Morphologically were found simple and composite grains. Cathodoluminescence
imaging shows that most zircon grains have concentric oscillatory zoning, irregular zoning and
are banded or unzoned. A total of 153 U-Th-Pb LA-ICP-MS analyses were performed on
detrital zircon grains. They show U content ranging from 15 to 6158 ppm. A population with 51
grains yielding U-Pb ages with 90-110% concordance (Fig. 8b) is dominated by Precambrian
ages (64%): Neoproterozoic (37%; c. 801-551 Ma), Paleoproterozoic (25%; c. 2.4-1.6 Ga) and
Neorchean (2%, c. 2.5 Ga). The Paleozoic grains (36%) are Carboniferous (20%; c. 359-303
Ma), Late Devonian (14%; c. 378-362 Ma), and Early Ordovician (2%; c. 447 Ma). The
youngest two grains (303±4 Ma; Kasimovian-Gzhelian) are slightly younger than the
sedimentary age inferred from biostratigraphic constraints (Middle Moscovian to Kasimovian;
Lemos de Sousa and Wagner, 1983; Machado et al., 2012; Lopes et al., 2014).
Sample SS-1 represents a very-coarse grained sandstone consisting of rounded-to-subangular
mono- and polycrystalline quartz, feldspar and muscovite grains, and a wide variety of
lithoclasts (chert, phyllite, rhyolite, siltstone and sandstone; Fig. 5h) . Zircon grains are rounded
to subangular, stubby and elongated prisms (less than 280 μm in diameter). The zircon
population includes simple grains with oscillatory concentric, banded and sector zoning, and
composite grains with cores with distinct internal morphologies surrounded by variable width
rims. A total of 150 U-Th-Pb LA-ICP-MS analyses performed on detrital zircon grains yielded
U content ranging from 24 to 9819 ppm. A group of 71 grains yielding U-Pb ages with 90-
110% concordance are dominated by Paleozoic ages (82%), predominantly made up of
Carboniferous (49%; c. 358-315 Ma) and Devonian (25%; c. 389-359 Ma), and a few Late
Ordovician-Silurian (5%; c. 434, 429 and 425 Ma) and Cambrian (3%; c. 533 and 491 Ma)
grains (Fig. 8b). The Precambrian grains (18%) are Neoproterozoic (10%; c. 702-542 Ma),
Paleoproterozoic (4%; c. 2.1-1.6 Ga), Mesoproterozoic (3%, c. 1.4 and 1.6 Ga) and Neorchean
(1%, c. 2.8 Ga). The youngest zircon population (N = 3; c. 319-315 Ma) suggest a Bashkirian
maximum depositional age, which is slightly older than the sedimentary age inferred from
biostratigraphic constraints (Middle Moscovian to Kasimovian; Lemos de Sousa and Wagner,
1983; Machado et al., 2012; Lopes et al., 2014).

**5. K-S test and MDS analysis: results**

We recognize that the representativeness of the detrital zircon grains of samples TM-1 and CB is not the most recommended for provenance analysis. However, we consider important to present a preliminary comparison of detrital zircon populations of Visean marine siliciclastic rocks from the Toca da Moura and Cabrela volcano-sedimentary complexes because it makes us suspect of variability in the sources. A table showing the K-S results (referred to as Fig. S1 throughout the text) can be found in the supplementary data repository (available online at _ link to be given by SOLID EARTH- ).

The K-S test performed on the Santa Susana sandstones show that the detrital zircon populations of sample SS-2 (lower member) and SS upper member (i.e. includes samples StSz2 and StSz4 from Dinis et al., 2018) are 'not significantly different' (all ages- P-value = 0.169; pre-Carboniferous ages- P-value = 0.879) at the 5% confidence level (Fig. S1). A comparison of samples SS-1 and SS-2 reveals that they are "significantly different" (P-value ≤ 0.01). Unlike sample SS-2, the sample SS-1 detrital zircon population is "significantly different" (P-value < 0.01) from the SS upper population (Fig. S1), indicating that they derived from distinct sources. Besides this, sample SS-1 is much closer to that of the SS upper (D-value = 0.323), and more distant from sample SS-2 (D-value = 0.465) as regards the distance between cumulative probability curves (Fig. 8c).

In Figure 9a, the MDS diagram produced with all ages shows sample SS-1 adjacent to Cabrela and Mértola siliciclastic rocks, while sample SS-2 is near the Mira, Santa Iria and Represa detrital zircon populations. In the MDS diagram for pre-Carboniferous ages, sample SS-2 is juxtaposed with sample TM-3, and closest to the Mira, Phyllite-Quartzite and Tercenas formations (Fig. 9b) suggesting likely sources. Nevertheless, the probable contribution to SS-2 samples of sediment derived from the oldest siliciclastic rocks from the PLZ and SPZ (i.e. Pulo do Lobo, Gafo, Ribeira de Limas, Atalaia and Ronquillo formations), and OMZ sources cannot be excluded. Their detrital zircon populations are 'not enough significantly different' (all ages- P-value = 0.003), and 'not significantly different' (pre-Carboniferous ages- P-value = 0.113-0.165) at the 5% confidence level (Fig. S1). This similarity is also illustrated in the approximation between SS-2, P-G-R-A-R and OMZ populations in the MDS diagrams (Figs. 9a, b).

K-S test results for the comparison between samples SS-2 and TM-3 indicate that they present 'not significantly different' detrital zircon populations (all ages- P-value = 0.399; pre-Carboniferous ages- P-value = 0.0.411) at the 5% confidence level (Fig. S1). Furthermore, their cumulative probability curves are much closer (Fig. 8d): D-values are 0.195 (all ages) and 0.203 (pre-Carboniferous ages) (Fig. S1). The close relationship of the two detrital zircon populations suggests that the Toca da Moura volcano-sedimentary complex directly supplied sediment to the Santa Susana basin. However, the relationship described above does not extend to the entire

Santa Susana basin since sample SS-1 presents a greater degree of similarity with the Cabrela
detrital zircon population as regards the proximity between cumulative probability curves (Fig.
8d) and MDS diagrams (Figs. 9a, b).
In addition, Cabrela siliciclastic rocks are 'significantly different' at the 5% confidence level
from sample TM-3 (P-values < 0.01) as regards the significant distance between them on the
MDS diagram (Figs. 9a, b), and the significant distance between cumulative curves (Fig. 8d),
with a D-value interval of 0.712-0.731 (Fig. S1). The difference found in the detrital zircon
populations suggests that Cabrela and Toca da Moura siliciclastic rocks probably derived from
different sources.
As result of the K-S test and MDS analyisis, the Horta da Torre Formation is 'significantly
different' (Fig. S1), and is clearly separate (Figs. 9a, b) from all the other detrital zircon
populations, ruling out the possibility of it being a source for the Toca da Moura and Cabrela
volcano-sedimentary complexes or Santa Susana Formation siliciclastic rocks.

**6. Discussion**
**6.1. Chronostratigraphic framework**
The geochronological data presented in the present study provide the basis for the first
chronostratigraphic record for the Carboniferous basins of the Santa Susana-São Cristovão
region (SW Iberia). Dating of silicic volcanic rocks interbedded in the Toca da Moura and
Cabrela volcano-sedimentary complexes constrain an interval of felsic magmatism to occurring
from c. 335 Ma to 331 Ma (Visean; Fig. 6), complementing currently-available biostratigraphic
information for Toca da Moura and Cabrela siliciclastic rocks (Pereira et al., 2006; Lopes et al.,
2014). U-Pb ages of the youngest detrital zircon grains from the siliciclastic rocks of the Toca
da Moura and Cabrela volcano-sedimentary complexes (TM-3 and CB, respectively; Fig. 8a)
provide maximum age constraints for these marine deposits. Their maximum depositional ages
(c. 351-348 Ma; Tournaisian) are slightly older than currently-available biostratigraphic ages
(Pereira et al., 2006; Lopes et al., 2014), but provide confirmation that both marine deposits are
broadly contemporaneous.
Furthermore, the best estimate of the crystallization age of the Baleizão silicic intrusion
provides a minimum age of 318 ± 2Ma (Bashkirian; Fig. 7a) for the intra-Carboniferous
unconformity. Zircon extracted from a pebble of granite found in a Santa Susana conglomerate
yielded a crystallization age of c. 303 Ma for plutonic rock (Fig. 7b). This age estimate overlaps
the age interval of c. 305-303 Ma (i.e. the maximum depositional age range) obtained for the
youngest population of detrital zircon grains from sandstone of the upper member (Dinis et al.,
2018), complementing the currently-available biostratigraphic information for the Santa Susana
Formation (Machado et al., 2012; Lopes et al., 2014). Given the findings described above, a
stratigraphic interval of approximately 13-17 Ma can be established for the intra-Carboniferous
unconformity, marking a change in depositional environment from marine to terrestrial in the
OMZ. Basin-drainage and infill patterns most probably changed due to rapid uplift of the
Variscan-Appalachian orogenic belt, active during the waning stages of Laurussia-Gondwana
collision (i.e. Late Carboniferous).

### 6.2. Provenance and evolutionary model

An initial important finding provides evidence that they derived from different sources. The
TM-3 population presents 64% Precambrian detrital zircon grains, while the CB population
contains only 10% (Fig. 8a). Toca da Moura siliciclastic rocks have a greater affinity with the
Phyllite-Quartzite, Tercenas, Santa Iria and Represa formations (Fig. 9), indicating that detrital
zircon populations were reproduced faithfully in SPZ and PLZ (Laurussian-type) sources. A
contribution from the oldest siliciclastic sequences of PLZ (Pulo do Lobo, Atalaia, Gafo and
Ribeira de Limas formations) and OMZ (Gondwanan-type) sources cannot be ruled out for
sample TM-3 (Fig. 9). The number of Late-Middle Devonian zircon grains in sample TM-3
(6%) is smaller than that of the CB population (68%) (Fig. 8a), suggesting that Cabrela
siliciclastic rocks were most likely derived largely from a Devonian source consistent with a
limited contribution from recycled ancient rocks (Pereira et al., 2012a). This indicates that the
origin of the Visean Toca da Moura and Cabrela basins is most likely more closely linked to
sources located in the SPZ and PLZ (Laurussian-type) than in the OMZ (Gondwanan-type). The
evidence in the Visean Toca da Moura basin for dissection of the inactive Devonian magmatic
arc and the erosion of its plutonic roots, together with the recycling of the PLZ and SPZ
Frasnian-Tournaisian siliciclastic sequences and OMZ basement rocks, differs from the
evidence in the Cabrela basin. The significance of the involvement of distinct sources is that
part of the region located on the boundary between the OMZ- PLZ and the SPZ (SW Iberia) was
subjected to uplift while the remaining part underwent flexural subsidence. A similar tectonic
setting has been put forward as an explanation for differences in stratigraphy found in the
Pedroches syn-orogenic basin located along the OMZ-Central Iberian Zone boundary
(Armendáriz et al., 2008, and references therein) (Fig. 1).
Over the past four decades, different models have emerged to explain the geodynamic evolution
of SW Iberia, with the subduction polarity being widely discussed (Quesada et al., 1994; Castro
et al., 1996; Ribeiro et al., 2007; Pin et al., 2008; Simancas et al., 2009; Braid et al., 2011;
Pérez-Cáceres et al. 2015a; Díez Fernández et al., 2016; Pereira et al., 2017a). Although in
many paleogeographic reconstructions for the Devonian, Iberia is flanked by the Rheic and
Paleotethys oceans (Stampfli et al., 2002, 2013; Cocks and Torsvik, 2006; Stampfli and Kozur,
2006; Torsvik et al., 2012; Arenas et al., 2014; von Raumer et al., 2016), solely the subduction
of Rheic Ocean is considered in the present geodynamic models for Iberia. This geodynamic
model of a single ocean has been the trigger of numerous discussions about whether the active
magmatic arc was located in Laurussia or Gondwana margins. Our challenging recent proposal
considers that SW Iberia geodynamic evolution could have been linked to the closure of these
two oceanic basins (Pereira et al., 2020). The main assumption that we must assume is that SPZ
and PLZ (Laurussian-side) and OMZ (Gondwana-side) have experienced different and
independent geodynamic evolutions before they were juxtaposed by the Variscan sinistral
orogen-parallel motion (Pérez-Cáceres et al., 2015b). As illustrated in Figure 10a, the
subduction of the Rheic Ocean floor beneath the Laurussian margin during the Late Devonian
(Pérez-Cáceres et al., 2015a; Pereira et al., 2017a, and references therein) caused the onset of
the Rheic magmatic arc in the Meguma terrane and related synorogenic basins. A slab rollback
mechanism similar to the one that caused the opening of the Okinawa trough behind the
Ryukyu-type subduction in the Pacific Ocean (Yamaji, 2003; Boutelier and Cruden, 2013)
could explain the lithosphere extension in the Laurussian-side and the Late Devonian
siliciclastic sedimentation in SPZ and PLZ (Pereira et al., 2017a). The Laurussian active margin
was progressively accreted to the Gondwana passive margin facing the Rheic Ocean during the
Late Devonian (Fig. 10a). A nappe stack was gradually emplaced in the Gondwana margin as a
consequence of ongoing continental collision (Pérez-Cáceres et al., 2015a; Díez Fernández and
Arenas, 2015, and references therein; Díez Fernández et al., 2016) and orogenic gravitational
collapse (Dias da Silva et al., 2020). Following the closure of the Rheic, Tournaisian, magmatic
episodes were associated with lithospheric extension in the Laurussian (Pyrite belt volcano-
sedimentary complex and Gil Márquez pluton) and Gondwanan (Beja igneous complex)
margins. A mechanism of steepening and break-off of the Rheic Ocean slab beneath the
Laurussian margin, as the modern analog of Eastern Anatolia Alpine-Hymalaya collisional
mountain belt (Sengor et al., 2003; Keskin, 2007), possibly was the main the reason for the
Early carboniferous magma generation in the SPZ (Pin et al. 2008), simultaneously the Meguma
terrane experienced rapid uplift and terrestrial sedimentation (Fig. 10b1). At the same time, in
the Gondwanan-side, the upwelling of the asthenosphere could have triggered partial melting of
crustal materials, and lithosphere extension (Pereira et al., 2009; 2012b), creating the right
conditions for the onset of gneiss domes in the OMZ (Dias da Silva et al., 2018). The
emplacement of voluminous subduction-related magmas (Santos et al., 1990; Castro et al.,
1996; Jesus et al., 2007; Pin et al., 2008; Lima et al., 2012; Pereira et al., 2007, 2015a; Moita et
al., 2009, 2015), including some with boninitic (Castro et al., 1999) and adakitic (Lima et al.,
2013) affinities,  was coeval with flexural subsidence, marine sedimentation and volcanism
(Toca da Moura, Cabrela, Los Pedroches basins) (Fig. 10b2). The Early Carboniferous thermal
anomaly recorded in the OMZ has been interpreted to result from the emplacement in the
middle crust of a large volume of mantle plume-related magmas (Simancas et al., 2006). Other
studies have suggested that voluminous Early Carboniferous magmatism could have resulted
from the subduction of an oceanic ridge, creating a slab window beneath the OMZ (Castro et al.,

1996, 1999; Diaz-Azpiroz et al., 2006). This model uses the Chile ridge that plunges beneath
South America Plate in Patagonia (Breitsprecher and Thorkelson, 2009) as modern analog. Our
model assumes the subduction of a ridge of the Paleotethys Ocean lithosphere beneath the
Gondwana margin (Fig. 10b2), instead of the Rheic Ocean lithosphere, as a hypothesis to be
further explored since the magmatic activity has extended to the Serpukhovian and Bashkirian
in the OMZ (Pavia pluton, Valencia del Ventoso plutonic complex and Baleizão porphyry).
Simultaneously with the putative subduction of the Paleotethys Ocean ridge, other regions of
the Appalachian-Variscan orogenic belt, mostly in the Laurussian-side (Fig. 10b1), have
experienced an oblique collision, rapid uplift and terrestrial sedimentation (Pereira et al., 2020).
A second significant finding is that detrital zircon populations from the Santa Susana Formation
(samples SS-1 and SS-2) also show significant differences (Figs. 8 and 9). Basal conglomerate
(sample SS-2) presents a greater percentage of Precambrian grains (64%) than uppermost
sandstone (SS-1 sample; 28%), and presents a great degree of affinity with the detrital zircon
population of sample TM-3. Sample SS-2 presents a great degree of similarity with the detrital
zircon populations of overlying SS upper-member sandstones (samples StSz-2 and StSz-4;
Dinis et al., 2018) sampled as part of the same stratigraphic profile. SS-2 and SS upper-age
populations show a great degree of affinity (Fig. 9), suggesting that detrital zircon grains were
mainly derived from the erosion of the Toca da Moura volcano-sedimentary complex, the Santa
Iria and Represa formations (PLZ) and the Mira Formation (SPZ). However, regarding the
detrital zircon grains with pre-Carboniferous ages, additional contributions from other PLZ
(Pulo do Lobo, Atalaia, Gafo and Ribeira de Limas formations), SPZ (Brejeira, Phyllite-
Quartzite, Tercenas and Ronquillo formations) and OMZ sources cannot be ruled out (Figs. 9a,
c). The zircon age population of sample SS-1, which is distinct from the SS-2 population,
presents a great degree of affinity with the CB population, suggesting lateral changes in sources
during deposition of Santa Susana uppermost sandstones. The great degree of affinity of the SS-
1, Cabrela volcano-sedimentary complex, with Mértola Formation detrital zircon populations
suggests a close association between the two and a common source. Cabrela and Mértola
siliciclastic rocks may be regarded as the main source for sample SS-1 and an intermediate
sediment repository as they are derived from the erosion of a Devonian source partially
represented by the Cercal porphyries from the SPZ. As result of rapid uplift, the progressive
erosion of the Devonian magmatic arc (including its plutonic roots), and that of PLZ, SPZ and
OMZ rocks, is evidenced in the Santa Susana Formation. The volumetrically significant
contribution of Carboniferous sources to the Santa Susana basin fill confirms derivation from
the erosion of: i) Pyrite Belt volcanic rocks, and Phyllite-Quartzite, Tercenas, Mértola, Mira and
Brejeira siliciclastic rocks (SPZ); ii) the Santa Iria and Represas formations (PLZ); iii) Gil
Marquez granitic rocks and other plutons of the Sierra del Norte Batholith (SPZ and PLZ); iv)
the Beja igneous complex, which includes the Baleizão porphyries (OMZ), and Évora and Pavia

plutonic and high-grade metamorphic rocks (OMZ); and v) the Cabrela and Toca da Moura
volcanic-sedimentary complexes (OMZ) and Mértola turbidites (SPZ). From Late
Carboniferous to Early Permian, large-scale strike-slip motions have juxtaposed OMZ to PLZ
and SPZ (García-Navarro and Fernández, 2004; Pérez-Cáceres et al., 2015b), simultaneously
with the rapid uplift of Variscan orogenic belt (Fig. 10c). In Kasimovian-Ghzelian,
sedimentation probably occurred through the opening of the pull-apart terrestrial basin (Santa
Susana basin) related to the movement of major strike-slip faults (i.e. Porto-Tomar fault zone,
Pereira et al., 2010; Machado et al., 2012; Gútiérrez-Alonso et al., 2015) during the progressive
uplift and buckling of the linear Appalachian-Variscan orogenic belt (i.e. OMZ, PLZ and SPZ;
Fig. 10c), related to a change in the regional stress-field that produced the Greater Cantabrian
Orocline (Pastor-Galan et al., 2015). U-Pb dating of magmatic zircon extracted from a pebble of
granite (c. 303 Ma; Fig. 7b) found in a conglomerate of the Santa Susana Formation lower
member suggests provenance from the direct erosion of Permo-Carboniferous plutons (i.e.
original primary source), such as Santa Eulália-Monforte granitic and gabbro-dioritic rocks
(OMZ). This c. 303-297 Ma calc-alkaline plutonic suite is coeval with the Nisa-Albuquerque
and Los Pedroches batholiths, located on the OMZ-Central Iberian Zone boundary (Fig. 1),
probably representing magmatism related to the eastward-migration of the Paleotethyan arc
(Permo-Carboniferous Pyrennes plutonic and volcanic rocks; Pereira et al., 2014; Pereira et al.,
2015b, 2017b). The Permo-Carboniferous OMZ plutons were emplaced at shallow crustal levels
consistent with the low assimilation of country rocks and the sharp contacts, and therefore, they
may have experienced denudation shortly after its crystallization without being required
unrealistic uplift rates.

**7. Conclusions**
The main conclusions of this study are the following:
1. Visean marine deposition in the Santa Susana-São Cristovão and Cabrela regions is
constrained to the age interval of c. 335-331 Ma by the new U-Pb data for volcanic rocks
intercalated within siliciclastic rocks of the Toca da Moura and Cabrela volcano-sedimentary
complexes.
2. U-Pb dating of the Baleizão porphyry provides a minimum age of $318 \pm 2$ Ma (Bashkirian) for
the overlying intra-Carboniferous unconformity.
3. Visean siliciclastic rocks from the Cabrela and Toca de Moura volcano-sedimentary
complexes are derived from distinct sources, which probably include a Devonian continental
magmatic arc, and are likely to be more closely associated with the Laurussian-type sources
(SPZ and PLZ) than the Gondwanan-type sources (OMZ).
4. Terrestrial siliciclastic rocks from the Santa Susana Formation are probably the result of the
recycling of distinct sources associated with the SPZ, PLZ and OMZ.

5. The best estimate of crystallization of a granite pebble found in Santa Susana Formation conglomerate suggest a maximum depositional age of c. 303 Ma (Kasimovian-Gzhelian); together with the youngest U-Pb ages (< c. 318 Ma) of detrital zircon grains, these findings provide evidence of the denudation of primary crystalline sources during the rapid post-accretion/collision uplift of the Variscan orogenic belt in SW Iberia (i.e. Gondwanan- and Laurussian-type sources).

6. The intra-Carboniferous unconformity that separates the Toca da Moura volcano-complex and the Baleizão porphyry from the Santa Susana Formation indicates a notable time interval of approximately 13-17 Ma.

**Acknowledgements**

This work is a contribution to projects CGL2016-76438-P and PGC2018-096534-B-I00 (Spain), the ICT's Research Group 6- Lithosphere Dynamics (ICT-UID/GEO/04683/2019) and, IDL's Research Group 3- Solid Earth dynamics, hazards, and resources (Portuguese FCT). Í. Dias da Silva acknowledges financial support by SYNTHESIS3-ACCESS (DE-TAF-5798), FCT postdoctoral grant SFRH/BPD/99550/2014 and FCT-project UID/GEO/50019/2019-IDL. This is IBERSIMS publication number 71.

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

**Figure captions**
Figure 1: A- Inset with location of SW Iberia in the Iberian Variscan belt with regional
distribution of the main Paleozoic terranes: CIZ- Central Iberian Zone; CZ- Cantabrian Zone;
GTMZ- Galicia-Trás-os-Montes Zone; OMZ- Ossa-Morena Zone; PLZ- Pulo do Lobo Zone;
SPZ- South-Portuguese Zone and WALZ- West Asturian-Leonese Zone. B- Simplified
Geological Map of SW Iberia showing the South-Portuguese, Pulo do Lobo and Ossa-Morena
zones (Modified from Pereira et al. 2017a, 2019 and references therein; Quesada and Oliveira,

995     2019).


Figure 2: Simplified geological map and schematic stratigraphy of the Santa Susana-São
Cristovão region (Ossa-Morena Zone; Modified from Gonçalves and Carvalhosa, 1984;
Machado et al., 2012). Sampling locations of the Carboniferous sedimentary and igneous rocks
used for geochronology are indicated with yellow stars.

Figure 3: Photographs of the Carboniferous igneous rocks of the Santa Susana-São Cristovão
region: A- Baleizão porphyry intrusive contact (yellow arrow) with siliciclastic rocks of the
Toca da Moura volcano-sedimentary complex; B- Baleizão porphyry; C-D- Rhyolitic tuffs of
the Toca da Moura volcano-sedimentary complex; E- Volcanic breccia with fragments of
siltstone (black) and rhyolite (yellow) at the base of the silicic tuffs from the Toca da Moura
volcano-sedimentary complex; F- Silicic volcanic rock interbedded with siltstones of the
Cabrela volcano-sedimentary complex.

Figure 4: Photographs of the Carboniferous sedimentary rocks of the Santa Susana Formation
lower member: A- View of dipping meter-thick beds of medium-coarse grained sandstone
intercalated with conglomerate; B- Planar-bedded coarse-grained sandstone; C- Plant imprints
in sandstone; D- Conglomerate with cobbles and pebbles of granite (G), quartzite (Q), silicic
porphyry (SP) and mafic volcanic rock (M); E- Conglomerate with pebbles of rhyolite (R),
phyllite (P), felsic tuff (T) and quartzite (Q).

Figure 5: Petrographic images of the Carboniferous sedimentary and igneous rocks of the Santa
Susana-São Cristovão region: A- Rhyolitic-rhyodacitic tuff of the Toca da Moura volcano-
sedimentary complex showing quartz and feldspar phenocrysts enclosed in ash matrix; B-
Rhyolitic tuff showing flattened dark-brown millimeter-sized pumice and lithoclasts enclosed in
ash matrix; C- Porphyritic texture of the Baleizão rhyodacite-rhyolite characterized by quartz,
plagioclase, K-feldspar, biotite and amphibole phenocryst embedded in a fine-grained silicic
matrix; D- Cobble of fine-grained granite showing graphic intergrowths of quartz and alkali
feldspar, found in conglomerate from the Santa Susana Formation; E- Siltstone of the Toca da
Moura volcano-sedimentary complex mostly composed of quartz grains and a few grains of
plagioclase (P), tourmaline (T), and rock fragments (L); F- Siltstone of the Cabrela volcano-
sedimentary complex showing fining upwards grading and a slump-fold; G-H, Sandstones from
the Santa Susana Formation with high percentage of lithoclasts (L) and a few feldspar (F).

Figure 6: Concordia diagrams, weighted mean of $^{206}Pb/^{238}U$ ages of analyzed zircon grains
extracted from silicic volcanic rocks of the Toca da Moura and Cabrela volcano-sedimentary
complex.

Figure 7: Concordia diagrams, weighted mean of $^{206}Pb/^{238}U$ ages of analyzed zircon grains of:
A- the Baleizão porphyry and B- the cobble of granite found in conglomerate from the Santa
Susana Formation.

Figure 8: Pie diagrams and Kernel Density Estimation (KDE) with U-Pb detrital-zircon ages of
siliciclastic rocks from: A- the Toca da Moura (TM-3, this study) and Cabrela (CB: CBR-11,
this study; and OM-200, Pereira et al., 2012a) volcano-sedimentary complexes, and B- the Santa
Susana Formation (SS-1 and SS-2, this study; and SS Upper member, StSz2 and StSz4 from
Dinis et al., 2018); C-  U-Pb age cumulative frequency plots applied to the U-Pb ages (90-110%
concordance) of detrital zircon grains from the Toca da Moura and Cabrela volcano-
sedimentary complexes, and the Santa Susana Formation.

Figure 9: Multi-Dimensional Scaling diagrams (Vermeesch, 2018) applied to the U-Pb ages (90-
110% concordance) of detrital zircon grains from the Toca da Moura (TM-3) and Cabrela (CB)
volcano-sedimentary complexes, and the Santa Susana Formation (SS1, SS2, SS upper
member), and different potential sources:  OMZ (Linnemann et al. 2008; Pereira et al. 2008,
2012c), PLZ (Pereira et al. 2017a; Pérez Cácerez et al. 2017), SPZ (Braid et al. 2011; Pereira et
al., 2012a, 2014a; Rodrigues et al. 2014). Abbreviations: MT- Mértola Formation; MR- Mira
formation; BJ- Brejeira formation; PQ-TRC- Phyllite-Quartzite and Tercenas formations; SI-
REP- Santa Iria and Represa formations; P-G-R-A-R- Pulo do Lobo, Gafo, Ribeira de Lima,
Atalaia and Ronquillo formations; HT- Horta da Torre Formation.

Figure 10: Sketches showing inferred tectonic evolution and sedimentation recorded in SW
Iberia Carboniferous stratigraphy during Laurussian-Gondwana oblique collision (Modified
from Pereira et al., 2012b; 2020); A- Late Devonian; B1-B2- Early Carboniferous; C1-C2- Late
Carboniferous.

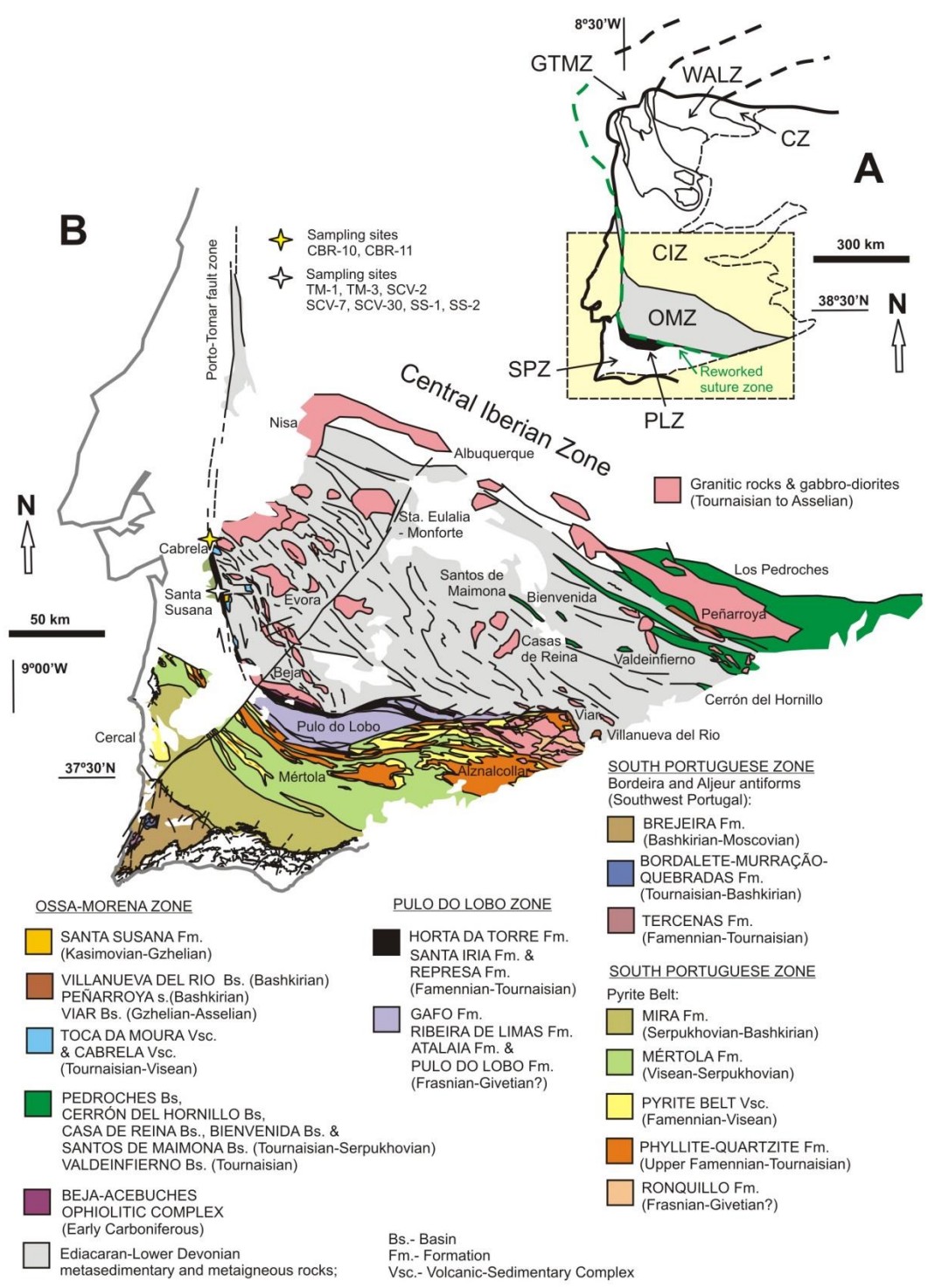

Figure 1

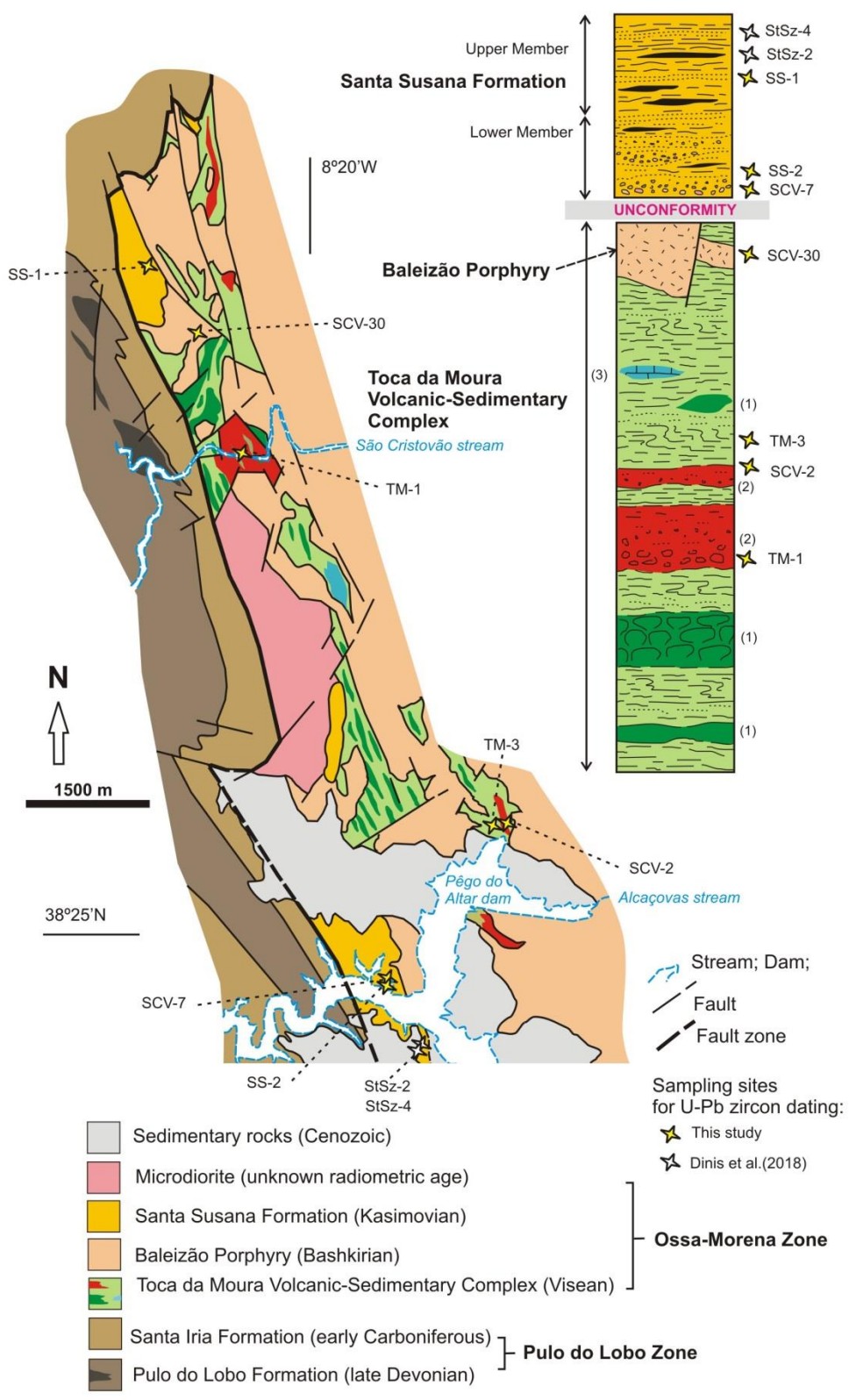

Figure 2

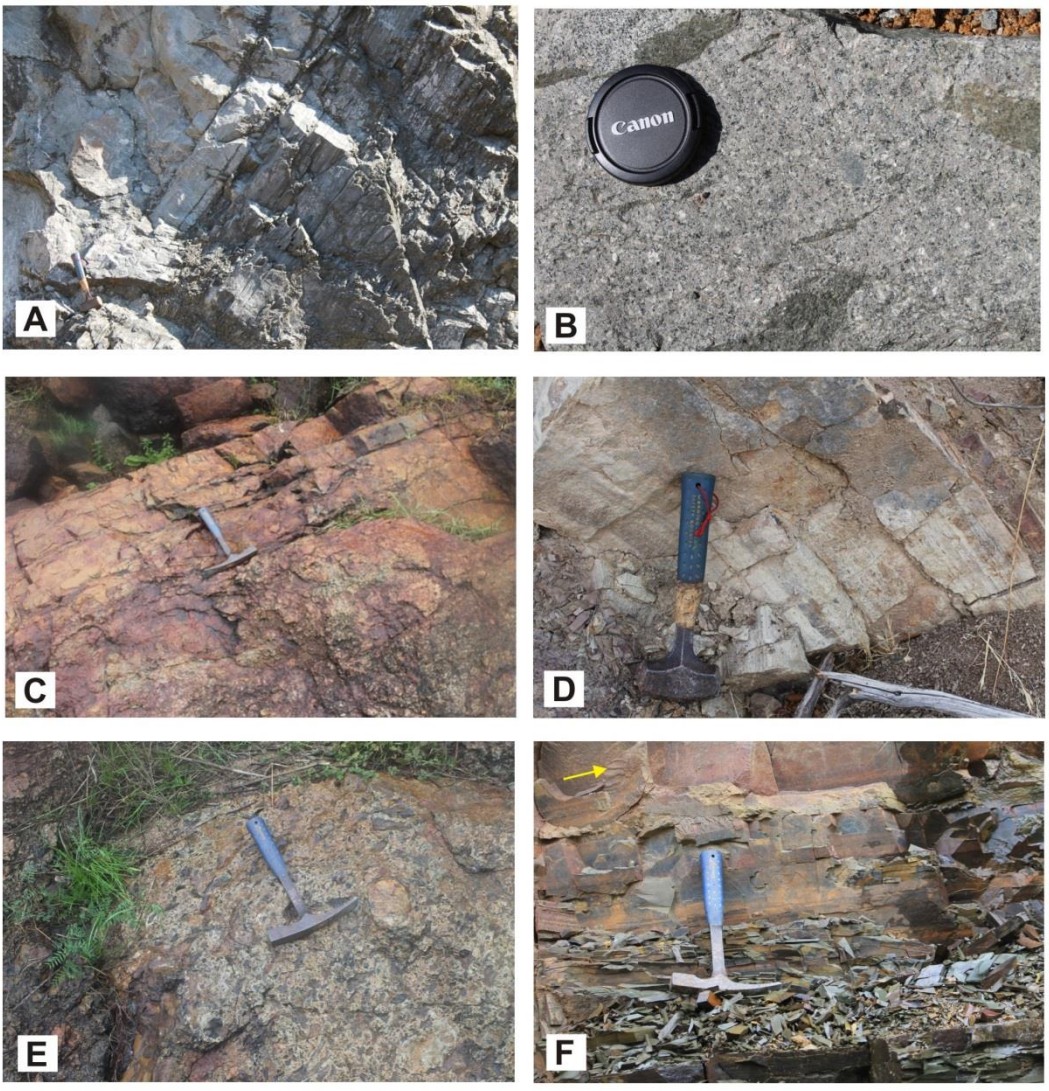

Figure 3

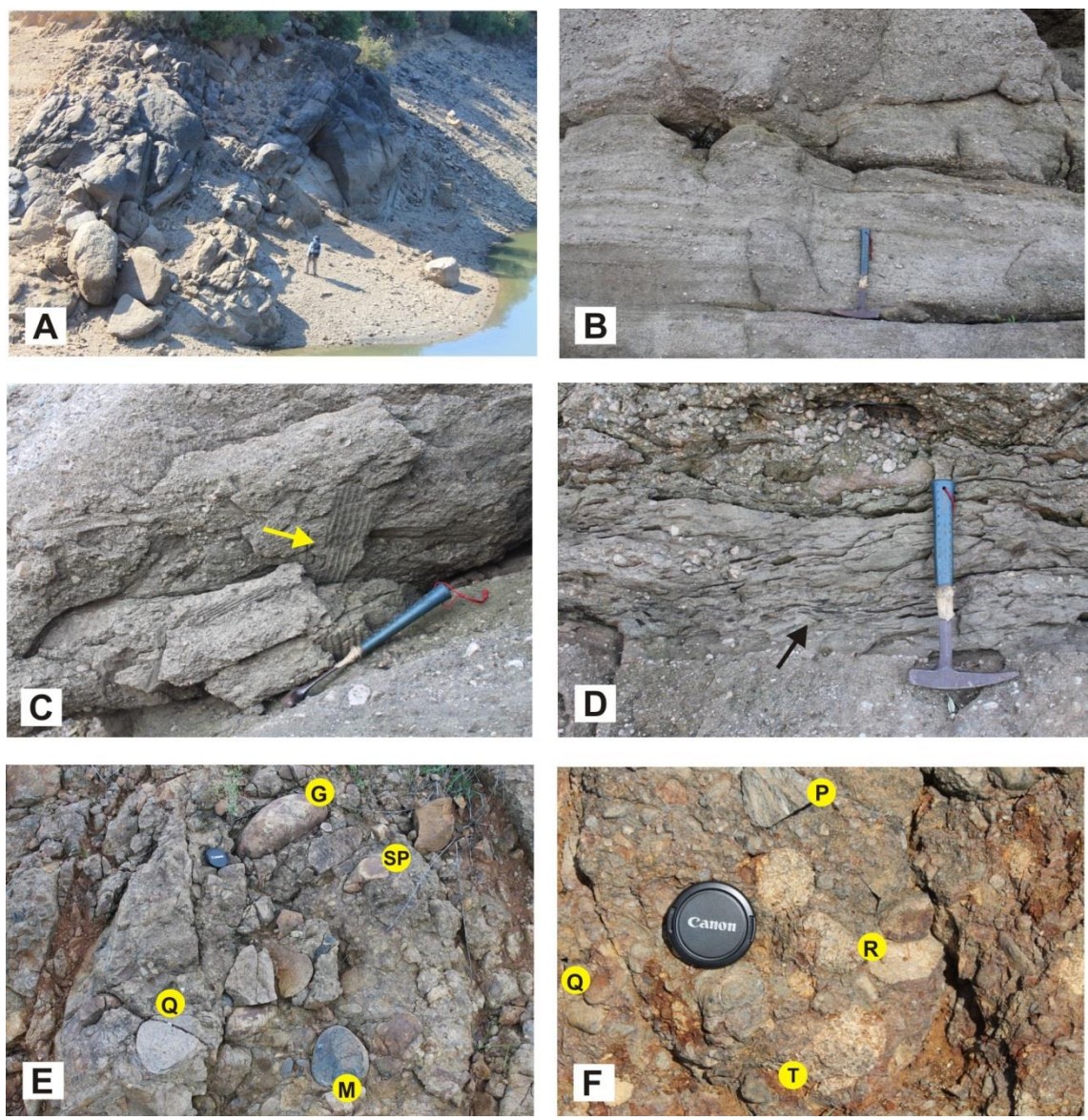

Figure 4

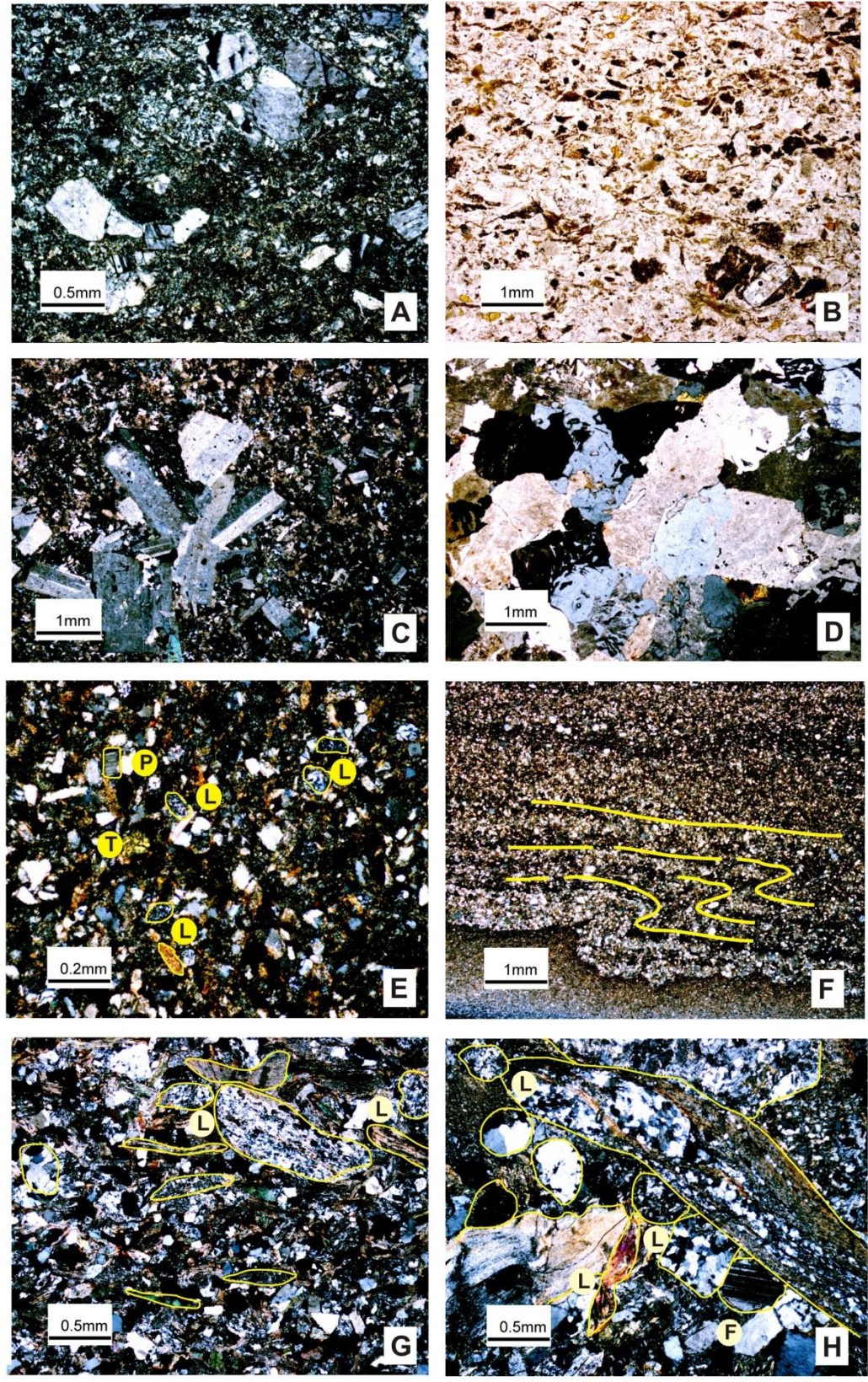

Figure 5

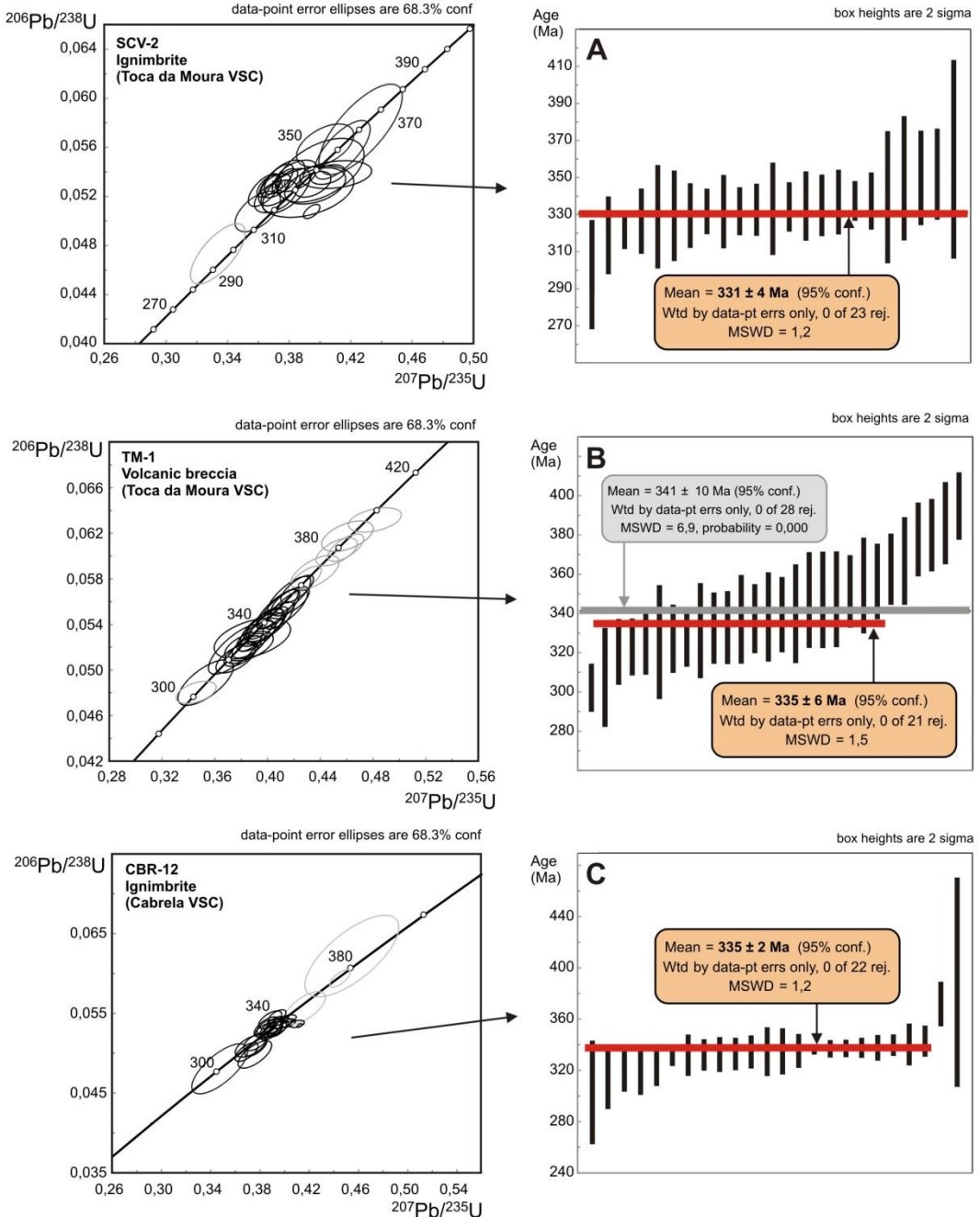

Figure 6

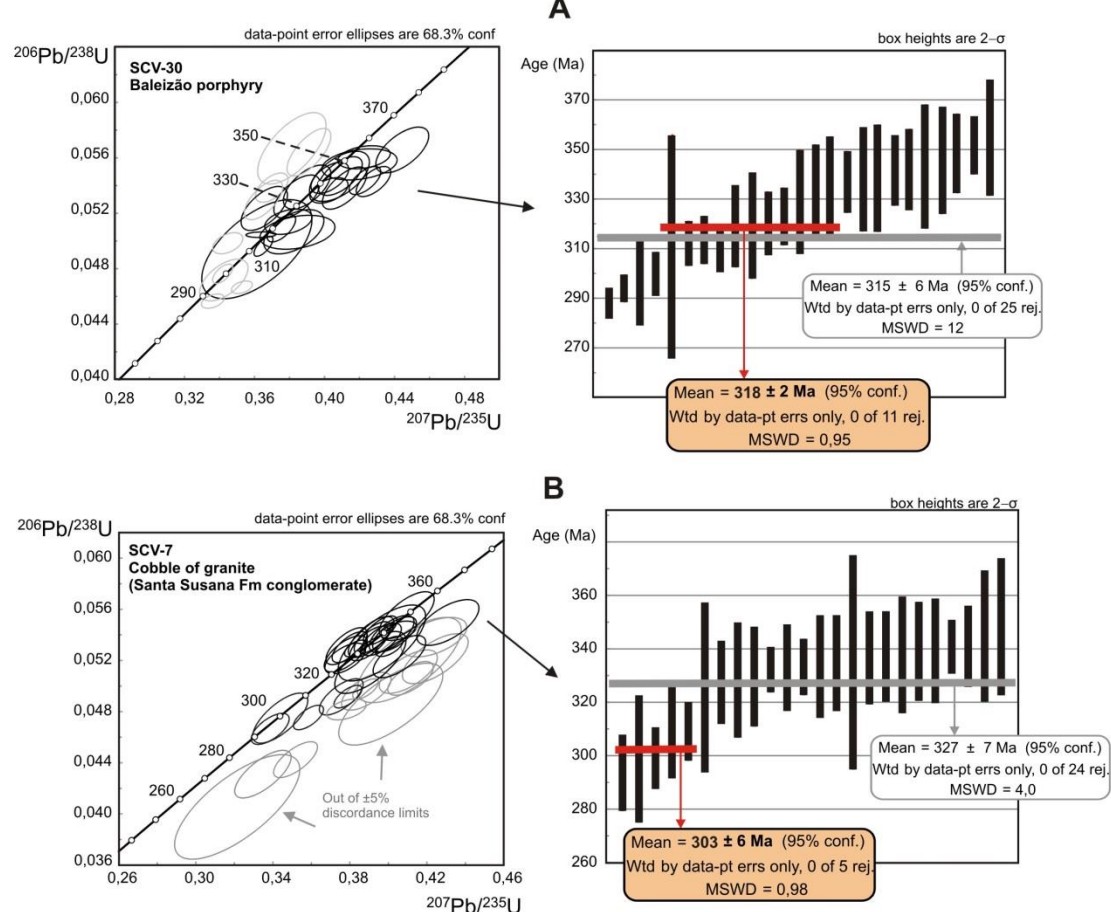

Figure 7

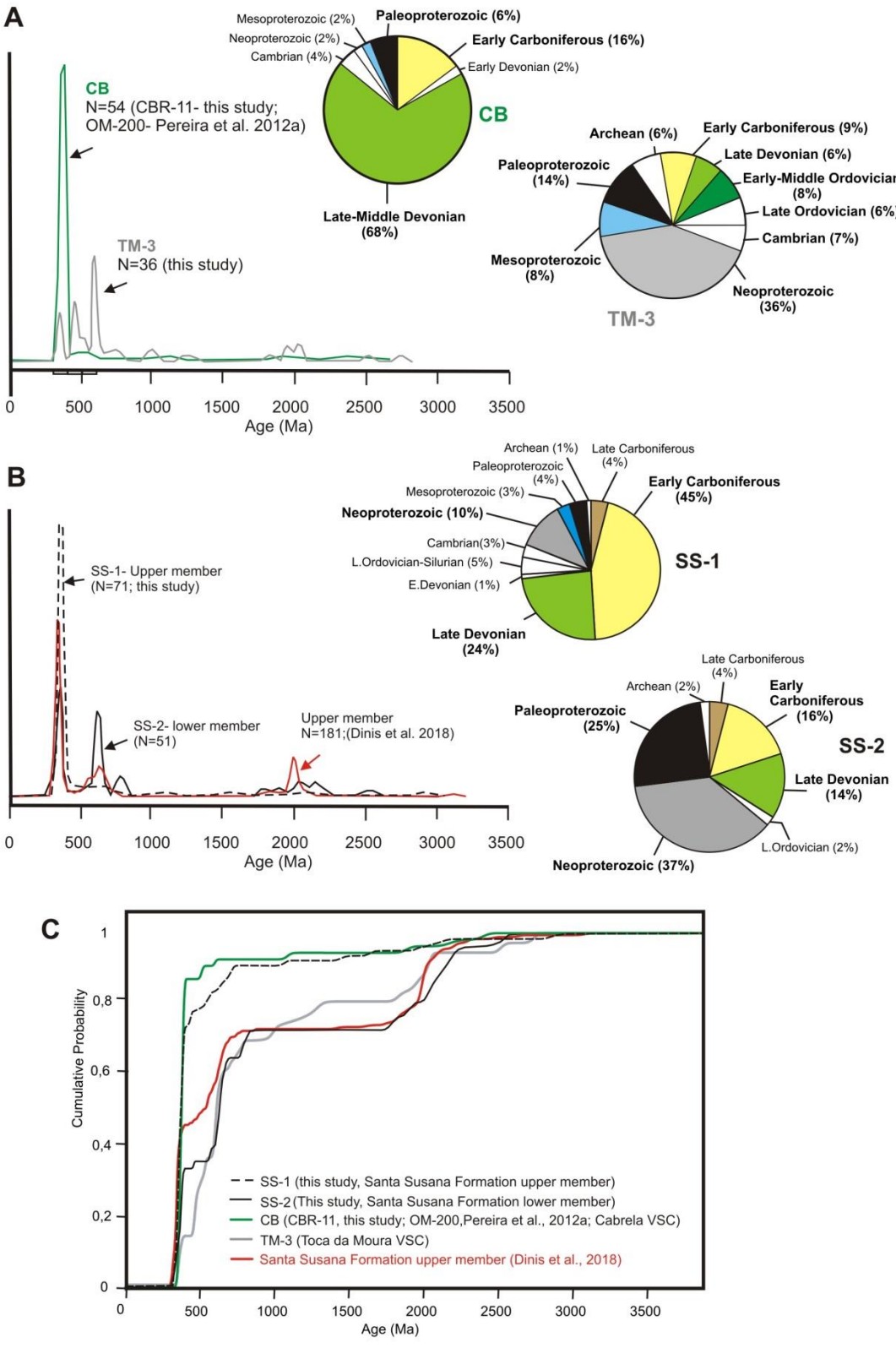

Figure 8

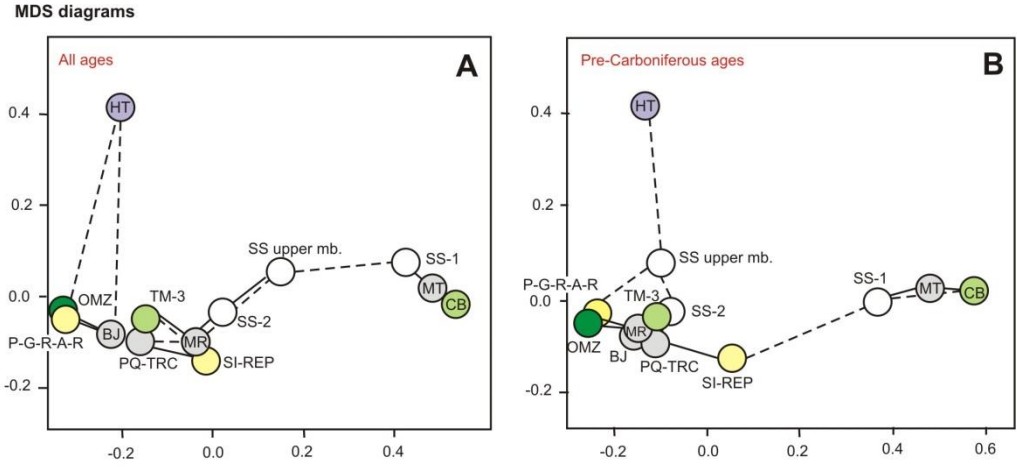

Figure 9

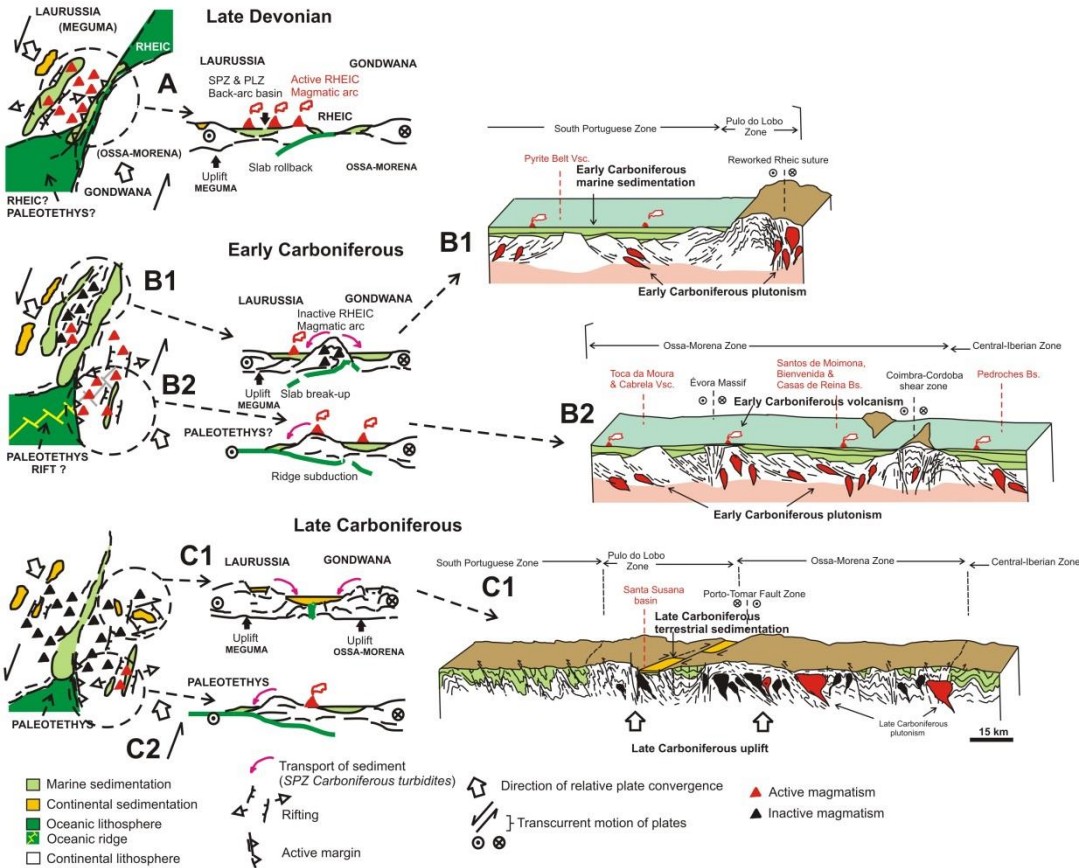

Figure 10