# Peer review of "Ossa-Morena Zone Carboniferous basins (SW Iberia)"

_Solid Earth, 2020_

## Referee Comment (RC1) · António Castro (Referee) · 18 Mar 2020

The paper supplies relevant interpretations on the provenance of Carboniferous basin materials of SW Iberia by means of analysing ages of detrital zircons. This study is a relevant contribution to understanding orogenic processes linked to basin evolution in one of the most complex zones of the European Variscan belt, the Ossa-Morena zone in SW Iberia. The manuscript is well written and clearly exposed in the most important principles and methodology. It can be improve by making some minor changes in several parts of the text. For instance, the Introduction can be improved by reorganizing the text and set the focus rather on the regional problem than the methodology. Thus,

[Figure]

Introduction must start on line 64, and the first paragraph (47-63) can move to item 3 (Methods). Lines 88-89 must go at the very beginning of the Introduction, as this a tribute volume. Because the paper is a regional contribution, the item "Geological setting" can be moved in part to the Introduction. A description of the sampled sedimentary units can be given in this second item after the introduction (like a material description). About the Discussion and interpretations. If there are implications of these new data on one of the most debated topics of SW Iberia, namely the polarity of subduction during closure of Rheic ocean, this must be discussed in this paper. Only few lines refer to this problem (446-461). For instance, if subduction was beneath the Laurussian margin, why the coeval arc magmatism is in the passive margin (Gondwana)? Subduction to north (beneath Ossa-Morena, the active Gondwana margin) is a more realistic interpretation according to structural and petrologic data.

———————————————————

---

## Author Comment (AC1) · 25 Mar 2020

Reply to the interactive comment of referee#1 (António Castro) on "Chronostratigraphic framework and provenance of the Ossa-Morena Zone Carboniferous basins (SW Iberia)" by M. Francisco Pereira et al. (Manuscript number se-2020-26).

First, the authors would like to thank the constructive comments made by referee#1 (António Castro), which contribute to improving our scientific work submitted for publication in Solid Earth Special issue. "The Iberian Massif in the frame of the European Variscan Belt".

[Figure]

Referee's comment 1: "The manuscript is well written and clearly exposed in the most important principles and methodology. It can be improved by making some minor changes in several parts of the text. For instance, the Introduction can be improved by reorganizing the text and set the focus rather on the regional problem than the methodology. Thus, Introduction must start on line 64, and the first paragraph (47-63) can move to item 3 (Methods). Lines 88-89 must go at the very beginning of the Introduction, as this a tribute volume. Because the paper is a regional contribution, the item "Geological setting" can be moved in part to the Introduction. A description of the sampled sedimentary units can be given in this second item after the introduction (like a material description)." Author's reply 1: "Comments on how to improve the organization of content by different sections (1- Introduction, 2- Geological Setting and 3- Rational and Analytical methods) are relevant and will be considered in the revised version that will be prepared for Solid Earth Special issue: "The Iberian Massif in the frame of the European Variscan Belt".

Referee's comment 2: "About the Discussion and interpretations. If there are implications of these new data on one of the most debated topics of SW Iberia, namely the polarity of subduction during the closure of the Rheic Ocean, this must be discussed in this paper. Only a few lines refer to this problem (446-461). For instance, if subduction was beneath the Laurussian margin, why the coeval arc magmatism is in the passive margin (Gondwana)? Subduction to the north (beneath Ossa-Morena, the active Gondwana margin) is a more realistic interpretation according to structural and petrologic data." Author's reply 2: "This geochronology study aims to establish the chronostratigraphic framework of the Carboniferous strata in the Santa Susana-São Cristóvão region (OMZ, SW Iberia) and to discuss their provenance. In section 6.2., the provenance of the Carboniferous strata is discussed based on a timeline related to the evolutionary model of the convergence between Laurussia and Gondwana. As referee #1 noted, our approach to the polarity of subduction during the Rheic Ocean closure is very brief. We chose to do so because this topic is discussed in another paper of ours (Pereira et al., in press) but, we will consider extending this discussion

in the revised version that will be prepared for Solid Earth Special issue "The Iberian Massif in the frame of the European Variscan Belt". We agree that referee #1 has highlighted a topic that has been much debated by geologists working in the Variscan orogen of SW Iberia. Over the past four decades, different models have emerged to explain the geodynamic evolution of SW Iberia, with the subduction polarity being widely discussed (Quesada et al., 1994; Castro et al., 1996; Ribeiro et al., 2007; Pin et al., 2008; Simancas et al., 2009; Braid et al., 2011; Pérez-Cáceres et al. 2015; Pereira et al., 2017a). This topic is quite complex and requires a careful analysis of the advances of scientific knowledge over the past four decades, which was based on different criteria that have been reassessed. As more data from geological mapping, structural geology, geochemistry, petrology, and geochronology were gathered, new hypotheses about the polarity of the subduction were suggested. The structural and petrological data (mentioned by referee #1) have been used extensively to justify the polarity of the subduction of oceanic lithosphere (Rheic Ocean) under Laurussia or under Gondwana (Pereira et al., 2017a and references therein). A major obstacle that exists, regarding the interpretation of the polarity of the subduction in SW Iberia, is that most models do not consider that Laurussia (i.e. PLZ and SPZ) will only have been juxtaposed to Gondwana (i.e. OMZ), regarding a geographic relationship comparable to the current one, in Late Carboniferous. That is, until then, from Late Devonian to Early Carboniferous, they probably have had a different and independent geodynamic evolution. Using a different perspective than the one followed by the previous models, we consider that the geodynamic evolution in SW Iberia may have been related to the closure of two oceanic basins that will have coexisted in the Devonian as suggested by paleogeographic reconstructions (Cocks and Torsvik, 2006; Stampfli and Kozur, 2006; Stampfli et al., 2013); Our recently proposed geodynamic model (Pereira et al., in press) admits: i) the closure of the Rheic Ocean in the Late Devonian under Laurussia, with the development of a magmatic arc (well-documented in the Meguma terrane, Nova Scotia) and synorogenic basins; this active margin of Laurussian was progressively accreted to the Gondwana passive margin facing the Rheic Ocean; at the same time, the Gondwana

passive margin facing the Paleotethys Ocean was developing (i.e. OMZ), immediately after the two ocean basins have coexisted ; ii) the onset of the closure of the Paleotethys Ocean in the Early Carboniferous under Gondwana (i.e. OMZ), and possibly in Laurussia regions facing this ocean (i.e. PLZ and SPZ) , where a magmatic arc (Gil Márquez, Beja and Évora plutons; Jesus et al., 2007; Gladney et al., 2014; Pereira et al., 2015a) and synorogenic basins (Mértola turbidites, Pyrite belt, Toca da Moura and Cabrela volcanic-sedimentary complexes) developed; high heat flow due to asthenospheric upwelling and extensive emplacement of Early Carboniferous igneous rocks in the OMZ (Pereira et al., 2009, 2015a) could have resulted from the subduction of an oceanic ridge (Castro et al., 1996; Díaz Azpiroz et al., 2006); and iii) in Late Carboniferous, large-scale transcurrent movements may have juxtaposed OMZ to PLZ and SPZ, simultaneously with the rapid uplift of Variscan orogeny; at the same time the Paleotethys subduction under Gondwana was responsible for the growth of a magmatic arc in Iberia until the Permian (Santa Eulália-Monforte, Nisa-Albuquerque, Los Pedroches plutons, NW and central Iberian plutons, and Pyrennes plutons and volcanism; Pereira et al., 2014; 2015b, 2017b).

References

Castro, A., Fernández, C., De la Rosa, J.D., Moreno-Ventas, I., El-Hmidi, H., El-Biad, M., Bergamín, J.F. and Sánchez, N.: Triple-junction migration during Paleozoic Plate convergence: the Aracena metamorphic belt, Hercynian Massif, Spain. Geologische Rundschau 85, 180-185, 1996.

Cocks, L.R.M. and Torsvik, T.H.: European geography in a global context from the Vendian to the end of the Palaeozoic, In: Gee, D. G. and Stephenson, R. A. (eds). European Lithosphere Dynamics. Geological Society, London, Memoirs, 32, 83-95, 2006.

Díaz Azpiroz, M., Fernández, C., Castro, A. and El-Biad, M.: Tectonometamorphic evolution of the Aracena metamorphic belt (SW Spain) resulting from ridge-trench interaction during Variscan plate convergence. Tectonics, 25, doi:10.1029/2004TC001742, 2006.

Gladney, E.R., Braid, J.A., Murphy, J.B., Quesada, C. and McFarlane, C.R.M.: U-Pb geochronology and petrology of the late Paleozoic Gil Márquez pluton: magmatism in the Variscan suture zone, southern Iberia, during continental collision and the amalgamation of Pangea. International Journal of Earth Sciences, 103, 1433-1451, 2014.

Pereira, M.F., Gama, C., Dias da Silva, Í., Fuenlabrada, J.M., Silva, J.B. and Medina, J.: Isotope geochemistry evidence for Laurussian-type sources of South-Portuguese Zone Carboniferous turbidites (Variscan orogeny). GSLSpecPub2019-163, in press.

Pereira, M.F., Castro, A., Chichorro, C., Fernández, C., Díaz-Alvarado, J., Martí, J. and Rodríguez, C.: Chronological link between deep-seated processes in magma chambers and eruptions: Permo-Carboniferous magmatism in the core of Pangaea (Southern Pyrenees). Gondwana Research 25, 290-308, 2014.

Stampfli, G.M., Hochard, C., Vérard, C., Wilhem, C. and von Raumer, J.: The formation of Pangea. Tectonophysics 593, 1-19, 2013.

Stampfli, G.M. and Kozur, H.: Europe from the Variscan to the Alpine cycles. In: Gee, D.G., Stephenson, R. (Eds.), European lithosphere dynamics: Geological Society, London, Memoir, 32, 57-82, 2006.

---

## Referee Comment (RC2) · Daniel Pastor-Galán (Referee) · 1 Apr 2020

The paper entitled "Chronostratigraphic framework and provenance of the Ossa-Morena Zone Carboniferous basins (SW Iberia)" co-authored by M. Francisco Pereira and collaborators presents new U-Pb zircon geochronology (LA-ICP-MS and SIMS) both from igneous and detrital rocks. The new dataset contains valuable absolute ages for several volcano-clastic and plutonic rocks, which in turn help to date the uplift and exhumation history of a basing and constrain the timing for a local-to-regional late Carboniferous unconformity.

In general terms, the paper is well written and easy to understand. I have only identified

a few typos and some sentences that I felt repetitive or adding superfluous information (see attached annotated PDF). In my opinion, the section 3 (Rationale and analytical methods) is unnecessarily long and tangled in a way that reads apologetic. You may consider rewording some paragraphs to enhance it and remark why this dataset is important and of general interest for the Paleozoic in Iberia and Europe (and I think it is).

From the studied datasets, I am happy with the study, statistical treatment and age interpretation of the igneous rocks. It is robust and well reasoned. However, you give and thoroughly describe U/Th but you never discuss them. Potential readers not familiar with zircon geochronology will wonder why is U/Th ration important at all and what is the meaning of those numbers you give and their average (does the average have any meaning considering some of the zircons are inherited?). I encourage you to discuss the meaning of the U/Th ratios and their implications to understand the origin of the zircons (metamorphic vs. igneous and the prospective igneous provenance of zircons - higher or lower temperatures). Otherwise, you may opt to not discuss at all the results, but once the results are there, I think it is interesting to give the whole picture.

I am a little less happy with the results of the detrital samples. I have noticed several minor but relevant issues (see the annotated PDF). Among them the relatively low number of analyzed zircons (some cases <40) in samples with too many peaks. In such cases every single zircon con turn easily the distribution. You are comparing these datasets with others to check their provenance, and with such short datasets the results can be misleading. I think the limitations of your new datasets should be, at least, mentioned in the paper. Also, treatment of the minimum depositional age, which sometimes is an average of several zircons (still don't get why, the youngest zircons in a detrital sample do not need to come all from the same rock and/or age) instead of giving the youngest concordant zircon with its uncertainty. Finally, I am unsure of how the K-S test gives any further or better information compared to MDS. MDS is basically the same but compares all the samples together and plots a really easy to understand graphics. Unless there are some relevant differences (not discussed in the txt right now, and I could not fine any) I recommend to move the K-S to the repository and treat it as a proof of concept instead.

Finally, As a curious note since I know it is not a major conclusion of this paper. I have problems to see how the subduction of the Paleotethys more than 600 km to the east (in present day coordinates and following Pereira's 2014; 2017a paleogeography) could cause arc magmatism in the sampled area. The average dip of the slab would be between 9ËŽ and 18ËŽ (assuming dehydration happens up to 200 km which is quite optimistic). Even a Puna style slab (with an initial steeper 30ËŽ slope to become later flat) dehydrates at some point 300-350 km far from the trench resulting in no more volcanism.

I have annotated other minor details in the annotated PDF.

I hope my comments are helpful to improve the paper.

Daniel Pastor-Galán

Please also note the supplement to this comment:
https://www.solid-earth-discuss.net/se-2020-26/se-2020-26-RC2-supplement.pdf

**Supplement:**

[revised manuscript text omitted]

Figure 1

[Figure]

[Figure]

[Figure]

Figure 2

[Figure]

[Figure]

Figure 3

[Figure]

[Figure]

Figure 4

[Figure]

[Figure]

[Figure]

Figure 5

[Figure]

[Figure]

Figure 6

[Figure]

Figure 7

[Figure]

[Figure]

[Figure]

Figure 8

[Figure]

[Figure]

Figure 9

[Figure]

[Figure]

[Figure]

Figure 10

---

## Author Comment (AC2) · 8 Apr 2020

We are grateful for the comments of referee#2 (Daniel Pastor-Galán) that are helpful to improve our paper.

Referee's comment 1: "From the studied datasets, I am happy with the study, statistical treatment and age interpretation of the igneous rocks. It is robust and well reasoned. However, you give and thoroughly describe U/Th but you never discuss them. Potential readers not familiar with zircon geochronology will wonder why is U/Th ration important at all and what is the meaning of those numbers you give and their average (does the average have any meaning considering some of the zircons are inherited?). I

encourage you to discuss the meaning of the U/Th ratios and their implications to understand the origin of the zircons (metamorphic vs. igneous and the prospective igneous provenance of zircons - higher or lower temperatures). Otherwise, you may opt to not discuss at all the results, but once the results are there, I think it is interesting to give the whole picture."

- Author's reply 1: "Most zircon grains from both samples SCV-2 and TM-1 of rhyolitic tuff show moderate to high Th/U (0.11-0.95) and 0.42 < Th/U average < 0.53, indicating an igneous origin (Heaman et al., 1990; Hanchar and Miller, 1993; Hoskin and Schaltegger, 2003), and precipitation from felsic-intermediate metaluminous sources; sample TM-1also have 16% of grains with very low Th/U (less than 0.1) typical of the zircon precipitated during the metamorphism or partial melting of a peraluminous rock in the presence of minerals with high Th/U (Williams and Claesson, 1987; Heaman et al., 1990; Williams, 2001; Rubatto 2002), and less than 3% with Th/U > 1, suggesting precipitation from a mafic source (Wang et al., 2011; Pereira et al., 2014). Sample SCV-30 of porphyritic rhyodacite-rhyolite have zircon grains that fall within the range 0.34< Th/U < 0.52 (Th/U average = 0.41), suggestive of crystallization from a relatively chemically homogeneous felsic-intermediate metaluminous source, close to the field of felsic peraluminous sources. Sample SCV-7 of granite include zircon grains with moderate to high Th/U (0.3-0.76) and Th/U average = 0.5 that are compositionally similar to those of sample SCV-30, indicating a comparable source. A simple comparison between igneous and sedimentary rocks illustrates that zircon grains of siliciclastic rocks show higher average Th/U ratios ranging from 0.66 to 0.81. However, if we complement the data from the sample CBR-11 with those from sample OM-200 (Pereira et al., 2012a), we find that the zircon grains fall within the range 0.2< Th/U < 1.69 and Th/U average = 0.5. Most zircon grains from sample TM-3 of siltstone show moderate to high Th/U (0.11-0.95), indicating precipitation from felsic-intermediate metaluminous sources; sample TM-3 also has 9% of grains with very low Th/U (less than 0.1) typical of the zircon precipitated during the metamorphism or partial melting of a peraluminous rock, and 26% with Th/U > 1, suggesting precipitation from a mafic source, which is

consistent with a high Th/U average = 0.75. The detrital zircon populations from samples SS-1 and SS-2 of sandstone are mostly represented by grains (81-84%) showing moderate to high Th/U (0.13-0.99) and 0.66 < Th/U average < 0.72, followed by a group of grains (14-15%) with Th/U > 1, probably derived from a mafic source, and a few grains (1-5%) with very low Th/U (less than 0.1) precipitated during the metamorphism or partial melting of a peraluminous rock. Nevertheless, we have to be aware that the concentrations of Th and U in zircon are primarily influenced by factors such as element availability within a reaction environment and partitioning behavior of these two actinide elements between zircon and co-existing minerals (i.e. high-Th minerals such as monazite and allanite), melts and fluids (Harley et al., 2007; Wan et al., 2011). The existence of monazite, epidote, and allanite, or concurrent growth of this high-Th phase may result in precipitation of zircon with a low Th concentration, and therefore a low Th/U ratio. Zircon precipitation from a partial melt before the crystallization of high-Th minerals may have higher Th concentration and therefore a moderate to high 0.15 < Th/U average < 3.2 (Carson et al. 2002; Kelly and Harley 2005), and therefore very low Th/U <0.1 metamorphic and recrystallized zircon could be not observed in high-grade metamorphic rocks (Wan et al., 2011). Given the evidence for variable zircon Th/U described above, Th/U values can only be used with caution and in concert with other, more integrative, chemical criteria to assess the origin of zircon within its textural context (Harley et al., 2007). This discussion is quite interesting but also rather complex, and so we prefer to leave it out of this work."

Referee's comment 2: "I am a little less happy with the results of the detrital samples. I have noticed several minor but relevant issues (see the annotated PDF). Among them the relatively low number of analyzed zircons (some cases <40) in samples with too many peaks. In such cases, every single zircon con turns easily the distribution. You are comparing these datasets with others to check their provenance, and with such short datasets, the results can be misleading. I think the limitations of your new datasets should be, at least, mentioned in the paper."

- Author's reply 2: "We agree with referee #2 when he says that the detrital zircon population of sample TM-3 is small (N = 36) instead of gathering the minimum of 60-100 grains generally used for this type of provenance studies (Vermeesch, 2004); on the other hand, it also states that we must make this limitation known to the reader. We agree with his recommendations and we will introduce in the final version a sentence referring that comparisons based on the proportions of ages need to be conducted with caution in such cases. Despite the difficulty in finding zircon grains in sample TM-3 of siltstone, 82 were analyzed but the results obtained reduced the concordant ages to less than half of the analyses performed. Kolmogorov-Smirnov test can be meaningful when comparing age populations with distinct sizes and for detecting differences using smallish datasets (N ≥20). It is important to note, that the Kolmogorov-Smirnov test is very sensitive to the proportions of ages present (Gehrels et al., 2012) so that this preliminary comparison between sample TM-3 and the other siliciclastic rocks should not be disregarded. K-S test highlights a significant difference concerning Precambrian grains; they are much less represented (8%) in the Cabrela volcano-sedimentary complex (10%; sample CB), than in the Toca da Moura volcano-sedimentary complex (64%; sample TM-3) and the overlying Santa Susana Formation (samples SS-1 and SS-2), suggesting changes in sediment sources during the Carboniferous. The minimum number of detrital zircon grains which is advisable to use in provenance analysis lead me to the study of Murphy et al. (2004) published in Geology using only two samples and very few detrital zircon grains (N = 23 and N = 28). They have discussed, in a preliminary way, the sources of the siliciclastic rocks from West Avalonia and Meguma terrains based on the presence or absence of Mesoproterozoic, Ordovician and Silurian detrital zircon grains. The new geochronological results were decisive to improve the paleogeographic models of the peri-Gondwanan realm of the Appalachian orogeny. The main differences in detrital zircon age populations described by Murphy et al. (2004) have been corroborated through new studies based on a larger number of detrital zircon ages (Waldron et al., 2009; White et al., 2018; Shellnutt et al., 2019).

Referee's comment 3: "Also, treatment of the minimum depositional age, which sometimes is an average of several zircons (still don't get why the youngest zircons in a detrital sample do not need to come all from the same rock and/or age) instead of giving the youngest concordant zircon with its uncertainty."

- Author's reply 3:" We only used the term "minimum depositional age" when using a zircon crystallization age of igneous rocks (sample SCV-30) and not detrital zircon ages from siliciclastic rocks."

Referee's comment 4: "Finally, I am unsure of how the K-S test gives any further or better information compared to MDS. MDS is basically the same but compares all the samples together and plots a really easy to understand graphics. Unless there are some relevant differences (not discussed in the txt right now, and I could not fine any) I recommend to move the K-S to the repository and treat it as a proof of concept instead."

- Author's reply 4:" We agree with referee #2. Kolmogorov-Smirnov (K-S) test results will be removed from Figure 9."

Referee's comment 5: "Finally, as a curious note since I know it is not a major con- clusion of this paper. I have problems to see how the subduction of the Paleotethys more than 600 km to the east (in present day coordinates and following Pereira's 2014; 2017a paleogeography) could cause arc magmatism in the sampled area. The aver- age dip of the slab would be between 9ËŽ and 18ËŽ (assuming dehydration happens up to 200 km which is quite optimistic). Even a Puna style slab (with an initial steeper 30ËŽ slope to become later flat) dehydrates at some point 300-350 km far from the trench resulting in no more volcanism."

- Author's reply 5:" When the paleogeographic map was drawn up with the represen- tation of the Paleotethys subduction (Pereira et al., 2015b, 2017b) our main purpose was to explain the late Carboniferous-early Permian magmatism in Iberia. In this paleo- geographic reconstruction, the late Carboniferous-early Permian OMZ and CIZ plutons, and the same age Pyrenees volcanism and plutonism (Pereira et al. 2014), are roughly located about 200-300 km far from the trench "what is quite optimistic". In Pereira et al.
(2015b, 2017b), the intention was to illustrate the Paleotethys subduction simplistically and preliminarily, far from considering the effect of kinematic variables on the thermal structure of the mantle wedge and the location of melting related to water transport in subduction zones with distinct slab dips (Grove et al., 2009). As we are now considering that the putative Paleotethys subduction started earlier in the early Carboniferous, the magmatism of this age found in the OMZ will have been displaced to the west by left-lateral strike-slip fault systems that were active until Moscovian (Pérez Cáceres et al., 2015) to early Permian (García-Navarro and Fernández, 2004). We will improve Figure 10 so that the tectonic model can be better understood by readers."

References Carson, C.J., Ague, J.J. and Coath, C.D.: U-Pb geochronology from Tonagh Island, East Antarctica: implications for the timing of ultra-high temperature metamorphism in the Napier Complex. Precambrian Research 116, 237-263, 2002.

García-Navarro, E, and Fernández, C.: Final stages of the Variscan Orogeny at the southern Iberian massif: Lateral extrusion and rotation of continental blocks. Tectonics 23, TC6001, DOI:10.1029/2004TC001646, 2004

Gehrels, G.: Detrital zircon U-Pb geochronology: current methods and new opportunities, Chapter 2, In: Tectonics of Sedimentary Basins: Recent Advances. Cathy Busby and Antonio Azor, eds, Blackwell Publishing Ltd., 47-62, 2012.

Grove, T.L., Till, C.B., Lev, E., Chatterjee, N., Médard, E.: Kinematic variables and water transport control the formation and location of arc volcanoes. Nature 459: 694-697, 2009.

Hanchar, J.M. and Miller, C.F.: Zircon zonation patterns as revealed by cathodoluminescence and backscattered electron images: implications for interpretation of complex crustal histories. Chemical Geology 110, 1-13, 1993.

Harley, S.L., Kelly, N.M., Moller, A.: Zircon behavior and the thermal histories of Mountain Chains. Elements 3, 25-30, 2007.

[Figure]

Heaman, L.M., Bowins, R. and Crocket, J.: The chemical composition of igneous zircon studies: implications for geochemical tracer studies. Geochimica et Cosmochimica Acta, 54, 1597–1607, 1990.

Hoskin, P.W.O. and Schaltegger, U: The composition of zircon and igneous and metamorphic petrogenesis, In: Hanchar, J.M., and Hoskin, P.W.O., eds., Zircon: Reviews in Mineralogy and Geochemistry, 53, 27–62, 2003.

Kelly, N.M. and Harley, S.L.: An integrated microtextural and chemical approach to zircon geochronology: refining the Archaean history of the Napier Complex, East Antarctica. Contributions to Mineralogy and Petrology 149: 57-84, 2005.

Murphy, J.B., Fernández-Suárez, J., Keppie, J.D., and Jeffries, T.E.: Contiguous rather than discrete Paleozoic histories for the Avalon and Meguma terranes based on detrital zircon data. Geology 32, 585-588, 2004.

Pereira, M.F., Gama, C. and Rodríguez, C.: Coeval interaction between magmas of contrasting composition (Late Carboniferous-Early Permian Santa Eulália-Monforte massif, Ossa-Morena Zone): field relationships and geochronological constraints. Geologica Acta 15, 409-428, 2017b.

Pereira, M.F., Castro, A., Fernández, C.: The inception of a Paleotethyan magmatic arc in Iberia. Geosciences Frontiers: 6, 297-306, 2015b.

Pereira, M.F., Castro, A., Chichorro, M., Fernández, C., Diaz-Alvarado, J., Martí, J. and Rodriguez, C.: Chronological link between deep-seated processes in magma chambers and eruptions: Permo- Carboniferous magmatism in the core of Pangaea (Southern Pyrenees). Gondwana Research 25, 290-308, 2014.

Pereira, M.F., Chichorro, M., Johnston, S., Gutiérrez-Alonso, G., Silva, J., Linnemann, U., Hofmann, M. and Drost, K.: The missing Rheic ocean magmatic arcs: provenance analysis of Late Paleozoic sedimentary clastic rocks of SW Iberia. Gondwana Research 22, 882-891, 2012a.

Pérez-Cáceres, I., Simancas, J.F., Martínez Poyatos, D., Azor, A. and González Lodeiro, F.: Oblique collision and deformation partitioning in the SW Iberian Variscides. Solid Earth, doi:10.5194/sed-7-3773-2015, 2015b.

Rubatto, D.: Zircon trace element geochemistry: partitioning with garnet and the link between U–Pb ages and metamorphism. Chemical Geology 184: 123-138, 2002.

Shellnutt, J.G., Owen, J.V., Yeh, M.-W., Dostal, J., and Nguyen, D.T.: Long-lived association between Avalonia and the Meguma terrane deduced from zircon geochronology of metasedimentary granulites. Scientific Reports, 9, 4065, 2019.

Vermeesch, P.: How many grains are needed for a provenance study? Earth and Planetary Science Letters, 224, 441-451, 2004.

Waldron, J.W.F., White, C.E., Barr, S.M., Simonetti, A. and Heaman, L.M.: Provenance of the Meguma terrane, Nova Scotia: Rifted margin of Early Paleozoic Gondwana. Canadian Journal of Earth Sciences, 46, 1-8, 2009.

Wan, Y., Liu, D., Dong, C., Liu, S., Wang, S., Yang, E.: U-Th-Pb behavior of zircons under high-grade metamorphic conditions: A case study of zircon dating of meta-diorite near Qixia, eastern Shandong. Geosciences Frontiers: 2, (2), 137-146, 2011.

Wang, X., Griffin, W.L., Chen, J., Huang, P., Li, X.: U and Th contents and Th/U ratios of zircon in felsic and mafic magmatic rocks: improved zircon-melt distribution coefficients. Acta Geologica Sinica, 85 (1), 164-174, 2011

White, C.E., Barr, S.M. and Linnemann, U.: U-Pb (zircon) ages and provenance of the White Rock Formation of the Rockville Notch Group, Meguma terrane, Nova Scotia, Canada: evidence for the "Sardian gap" and West African origin. Canadian Journal of Earth Science 55(6), 589-603, 2018.

Williams, I.S. and Claesson, S.: Isotopic evidence for the Precambrian provenance and Caledonian metamorphism of high-grade paragneisses from the Seve Nappes, Scandinavian Caledonides, II Ion microprobe zircon U–Th–Pb. Contributions to Mineralogy

and Petrology 97, 205-217, 1987.

Williams, I.S.: Response of detrital zircon and monazite, and their U-Pb isotopic systems, to regional metamorphism and host-rock partial melting, Cooma Complex, southeastern Australia. Australian Journal of Earth Sciences, 48, 557-580, 2001.

---

## Author Response (AR1)

Universidade de Évora, 9-5-2020

Dear Topic Editor of SOLID EARTH
Special Issue: The Iberian Massif in the frame of the European Variscan Belt
Dr. Emilio González-Clavijo

Subject:
Submission of revision manuscript se-2020-26 by M. Francisco Pereira (myself-corresponding author), Cristina Gama, Ícaro Dias da Silva, José B. Silva, Mandy Hofmann, Ulf Linnemann and Andreas Gartner.

Please find attached the electronic revised version of: "Chronostratigraphic framework and provenance of the Ossa-Morena Zone Carboniferous basins (SW Iberia)"".

This new version includes suggestions from reviewers #1 and #2 (changes marked in blue).
Figures 1, 3, 6, 7, 9 and 10 were improved and uploaded.
Tables 1 and 2 from supplementary material were changed.
Figure 9a was moved to supllementary material as Figure S1.

We hope that this new revised version could be accepted for publication in SOLID EARTH

Sincerely yours,

M. Francisco Pereira

Instituto de Ciências da Terra
Departamento de Geociências, Escola de Ciências e Tecnologia, Universidade de Évora
Colégio Luis Antonio Verney, Apartado 94, 7002-554 Evora, Portugal
Tel: 00 351 266 745301; Fax: 00 351 266 745397; E-mail: mpereira@uevora.pt

**Reviewer #1**

*Referee's comment 1: "The manuscript is well written and clearly exposed in the most important principles and methodology. It can be improved by making some minor changes in several parts of the text. For instance, the Introduction can be improved by reorganizing the text and set the focus rather on the regional problem than the methodology. Thus, Introduction must start on line 64, and the first paragraph (47-63) can move to item 3 (Methods). Lines 88-89 must go at the very beginning of the Introduction, as this a tribute volume.*

- We have considered very useful this suggestion of reviewer#1 and we change the text accordingly.

*Referee's comment 2: "Because the paper is a regional contribution, the item "Geological setting" can be moved in part to the Introduction. A description of the sampled sedimentary units can be given in this second item after the introduction (like a material description)."*

- This amendment suggested by reviewer#1 is not essential and therefore we have not adopted it.

*Referee's comment 3: "About the Discussion and interpretations. If there are implications of these new data on one of the most debated topics of SW Iberia, namely the polarity of subduction during the closure of the Rheic Ocean, this must be discussed in this paper. Only a few lines refer to this problem (446-461). For instance, if subduction was beneath the Laurussian margin, why the coeval arc magmatism is in the passive margin (Gondwana)? Subduction to the north (beneath Ossa-Morena, the active Gondwana margin) is a more realistic interpretation according to structural and petrologic data."*

- In order to satisfy the suggestions of reviewer#1, which we think are very pertinent, we extended the discussion of the geodynamic model and improved figure 10 to illustrate the text now presented.

**Reviewer #2**

Most of the notes presented by this reviewer were followed in the attached pdf that he made available; some of their observations coincide with their main recommendations that follow:

*Referee's comment 1: "From the studied datasets, I am happy with the study, statistical treatment and age interpretation of the igneous rocks. It is robust and well reasoned. However, you give and thoroughly describe U/Th but you never discuss them. Potential readers not familiar with zircon geochronology will wonder why is U/Th ration important at all and what is the meaning of those numbers you give and their average (does the average have any meaning considering some of the zircons are inherited?). I encourage you to discuss the meaning of the U/Th ratios and their implications to understand the origin of the zircons (metamorphic vs. igneous and the prospective igneous provenance of zircons - higher or lower temperatures). Otherwise, you may opt to not discuss at all the results, but once the results are there, I think it is interesting to give the whole picture."*

- After thinking about reviewer#2 considerations we decided to remove from the text the information related to the Th/U ratios of the zircon grains to avoid widening the discussion about the source of the melts from which they crystallized, whose theme is complex and quite controversial;

*Referee's comment 2: "I am a little less happy with the results of the detrital samples. I have noticed several minor but relevant issues (see the annotated PDF). Among them the relatively low number of analyzed zircons (some cases <40) in samples with too many peaks. In such cases, every single zircon con turns easily the distribution. You are comparing these datasets with others to check their provenance, and with such short datasets, the results can be misleading. I think the limitations of your new datasets should be, at least, mentioned in the paper."*

- We agree with the observation of reviewer#2 that the number of ages of detrital zircon grains used in the analysis of provenance does not comply with the minimum established by recent studies on the subject. However, being aware of the limitation that this brings us to the data discussion, we want to publish them as a preliminary approach. In this sense, we decide to write a statement in the text to alert the reader to our limitation, as suggested by reviewer#2. Still, in order to satisfy some doubts, presented by the reviewer in his annotated pdf.file, about the calculation of the crystallization age of the Baleizão porphyry we added more data to make the result more robust; We also gathered geochronology data from a new sample of volcanic rock from the Cabrela basin that allowed us to consolidate the age of Early Carboniferous volcanism.

*Referee's comment 3: "Also, treatment of the minimum depositional age, which sometimes is an average of several zircons (still don't get why the youngest zircons in a detrital sample do not need to come all from the same rock and/or age) instead of giving the youngest concordant zircon with its uncertainty."*
- We didn't find any reason for this criticism from reviewer#2, because we only applied the term "minimum depositional age" for referring zircon crystallization age of igneous rocks (sample SCV-30), and not for detrital zircon ages from siliciclastic rocks.

*Referee's comment 4: "Finally, I am unsure of how the K-S test gives any further or better information compared to MDS. MDS is basically the same but compares all the samples together and plots a really easy to understand graphics. Unless there are some relevant differences (not discussed in the txt right now, and I could not fine any) I recommend to move the K-S to the repository and treat it as a proof of concept instead."*

- Following this suggestion of reviewer#2 we move to supplementary material the tables with K-S results presented in Fig. 9a of the previous version;

*Referee's comment 5: "Finally, as a curious note since I know it is not a major conclusion of this paper. I have problems to see how the subduction of the Paleotethys more than 600 km to the east (in present day coordinates and following Pereira's 2014; 2017a paleogeography) could cause arc magmatism in the sampled area. The average dip of the slab would be between 9ËŽ and 18ËŽ (assuming dehydration happens up to 200 km which is quite optimistic). Even a Puna style slab (with an initial steeper 30ËŽ slope to become later flat) dehydrates at some point 300-350 km far from the trench resulting in no more volcanism."*

- As requested by the previous reviewer#1, this reviewer#2 also asked us to deepen the discussion of the evolutionary model and we did so.

---

## Author Response (AR2)

Universidade de Évora, 9-5-2020

Dear Topic Editor of SOLID EARTH
Special Issue: The Iberian Massif in the frame of the European Variscan Belt
Dr. Emilio González-Clavijo

Subject:
Submission of revision manuscript se-2020-26 by M. Francisco Pereira (myself-corresponding author), Cristina Gama, Ícaro Dias da Silva, José B. Silva, Mandy Hofmann, Ulf Linnemann and Andreas Gartner.

Please find attached the electronic revised version of: "Chronostratigraphic framework and provenance of the Ossa-Morena Zone Carboniferous basins (SW Iberia)", incorporating your last suggestions.

This new version includes two changes in the text:

- Lines 373-375: "A table showing the K-S results (referred to as Fig. S1 throughout the text) can be found in the supplementary data repository (available online at _ link to be given by SOLID EARTH- )."
-
- Lines 605-606: "This is IBERSIMS publication number 71."

I hope that everything is now ready to be accepted for publication in SOLID EARTH.

Sincerely yours,

M. Francisco Pereira

Instituto de Ciências da Terra
Departamento de Geociências, Escola de Ciências e Tecnologia, Universidade de Évora
Colégio Luis Antonio Verney, Apartado 94, 7002-554 Evora, Portugal
Tel: 00 351 266 745301; Fax: 00 351 266 745397; E-mail: mpereira@uevora.pt